# DeLLMa: Decision Making Under Uncertainty with Large Language Models

**Ollie Liu**\*, **Deqing Fu**\*, **Dani Yogatama, Willie Neiswanger**
Thomas Lord Department of Computer Science
University of Southern California
`me@ollieliu.com, {deqingfu, yogatama, neiswang}@usc.edu`

## Abstract

The potential of large language models (LLMs) as decision support tools is increasingly being explored in fields such as business, engineering, and medicine, which often face challenging tasks of *decision-making under uncertainty*. In this paper, we show that directly prompting LLMs on these types of decision-making problems can yield poor results, especially as the problem complexity increases. To aid in these tasks, we propose DeLLMa (Decision-making Large Language Model assistant), a framework designed to enhance decision-making accuracy in uncertain environments. DeLLMa involves a multi-step reasoning procedure that integrates recent best practices in scaling *inference-time reasoning*, drawing upon principles from decision theory and utility theory, to provide an accurate and human-auditable decision-making process. We validate our procedure on multiple realistic decision-making environments, demonstrating that DeLLMa can consistently enhance the decision-making performance of leading language models, and achieve up to a 40% increase in accuracy over competing methods. Additionally, we show how performance improves when scaling compute at test time, and carry out human evaluations to benchmark components of DeLLMa.

## 1 Introduction

Large language models (LLMs) are rapidly gaining traction across many domains due to their potential for automating and enhancing a broad spectrum of tasks (Bommasani et al., 2021; Bubeck et al., 2023). One important potential use is in *decision making under uncertainty*, i.e., deciding which action to take given some set of possibilities, properly factoring in user goals and uncertainty about the world. The ability to make good decisions under uncertainty holds broad relevance across high-stakes tasks in fields such as business, marketing, medicine, aeronautics, and logistics (Kochenderfer, 2015; Peterson, 2017)—and its value is not limited to organization-level decisions but extends to aiding individuals in making informed choices as well. The ability of LLMs to analyze large quantities of data makes them potentially well-suited for sophisticated decision support tools, and ensuring that these models give accurate, context-aware recommendations could significantly augment human decision-making capabilities.

However, optimal decision making under uncertainty is often challenging. For humans, there exist frameworks from decision theory and utility theory (developed in fields such as economics, statistics, and philosophy) to provide a structured approach for better decision-making (Von Neumann & Morgenstern, 1944; Luce & Raiffa, 1989; Berger, 2013). Research has consistently demonstrated that without these frameworks, human decision-making can often be highly irrational, swayed by biases and incomplete information (Bazerman & Moore, 2012). Similarly, making decisions with LLMs faces its own set of challenges. Issues include the tendency to fixate on specific explanations or information without adequately balancing all evidence, and the inability to effectively handle uncertainty, manage biases, or align with a user's goals and utilities (Ferrara, 2023; Benary et al., 2023). Our paper presents experiments that exemplify these issues.

Furthermore, beyond merely making rational decisions, it is crucial to understand *why* an LLM made a particular decision. This aids in building trust in the decision, assessing its quality, and improving any components that may lead to suboptimal outcomes. The ability to explain decisions and verify

---

\*Equal Contribution. Project website and code available at `https://dellma.github.io/`.

decision-making quality—which we refer to as *human auditability*—is essential for the practical application of LLMs to aid decision making in many real problems (Thirunavukarasu et al., 2023).

In this paper, our goal is to develop a framework that enables LLMs to make better decisions under uncertainty. Our aim is not only to enhance decision-making accuracy but also to allow human users to understand the rationale behind each decision. Drawing inspiration from prior work on multi-step reasoning like Chain-of-Thought (CoT) (Wei et al., 2022) and Tree-of-Thoughts (ToT) (Yao et al., 2023), in which compute is scaled at inference time, we design a procedure based on classical decision theory, originally designed for rational decision making under uncertainty by humans. Our approach involves three key steps: first, identify and forecast pertinent unknown variables given in-context information; second, elicit a utility function that aligns with the user's goals; and finally, use this utility function to identify the decision that maximizes expected utility. We call our proposed framework *DeLLMa*, short for Decision-making Large Language Model assistant.

We evaluate DeLLMa on real decision-making scenarios in agriculture and finance, and compare it against existing strategies for LLM decision-making, including zero-shot (direct) prompting, self-consistency (Wang et al., 2022), and CoT approaches. We find that DeLLMa significantly enhances decision-making accuracy, with improvements of up to a 40% increase in accuracy, particularly as the complexity and number of potential actions increases. Additionally, DeLLMa can consistently enhance performance across a variety of leading language models, and its structure allows us to understand the rationale behind each decision. In full, our contributions are:

- We introduce DeLLMa, a method for human-auditable LLM decision making under uncertainty, employing a multi-step reasoning process at inference time based on classical decision theory.
- We evaluate components of DeLLMa, including the calibration of its state forecasting approach and a human agreement study to assess its utility elicitation method.
- On realistic decision-making environments, we show that DeLLMa gives up to a 40% improvement in decision-making accuracy over competing methods, and yields consistent improvements when deployed across multiple leading LLMs.

## 2 RELATED WORK

Exemplified by OpenAI o1 (OpenAI, 2024), a combination of recent advances in scaling *inference-time reasoning*—such as task decomposition and structured search—has brought forth superhuman performances on deterministic reasoning tasks (Hendrycks et al., 2021; Chen et al., 2021); but we nonetheless find that they are insufficient for decision-making *under uncertainty*. DeLLMa offers a specialized inference-time solution that scales favorbly with *parallel sampling* (Snell et al., 2024), a key ingredient for performant utility elicitation and decision optimization. Below, we summarize prior works pertaining to the DeLLMa framework.

**LLMs for Decision Making.** Prior works have leveraged LLMs for optimizing blackbox functions (Yang et al., 2023; Nie et al., 2023; Shinn et al., 2024). These settings involve methods that make a substantial number of low-cost decisions (which do not incur a high price for suboptimality). Instead, we focus on single-step *expensive* decisions, particularly in the prescence of uncertainty, with a focus on the optimality of decisions. Additionally, a number of applied domains that involve decision making have started to explore LLM-based methods, such as in supply chain optimization (Li et al., 2023), medicine and health (Benary et al., 2023), and automated driving (Mao et al., 2023).

**Uncertainty in LLMs.** LLMs, without proper calibration, can be overly confident in their responses (Si et al., 2022). Such pitfalls make them unlikely to make reliable decisions under uncertainty. Prior work has aimed to solve this issue; one line of research involves asking LLMs for their own confidence, with or without additional finetuning (Kadavath et al., 2022; Lin et al., 2022; Mielke et al., 2022; Chen & Mueller, 2023; Tian et al., 2023; Xiong et al., 2023; Zeng et al., 2024). Referring to (Baan et al., 2023) for a detailed survey, many recent advances (Lin et al., 2023; Feng et al., 2025; Falck et al.; Wong et al., 2023) adopt a Bayesian inference framework to quantify and reason with uncertainties in LLMs. Other works have shown that tool usage (Ren et al., 2023), retrieval augmentation (Halawi et al., 2024), and model ensemble (Schoenegger et al., 2024) can improve calibration and forecasting capabilities of LLMs. Our framework can take advantage of these advances in LLM-based forecasting for improved decision making.

**Inference-time Reasoning in LLMs.** Many recent works have leveraged *inference time compute* to extend the computational power of language models (Merrill & Sabharwal, 2023; Sun et al.,

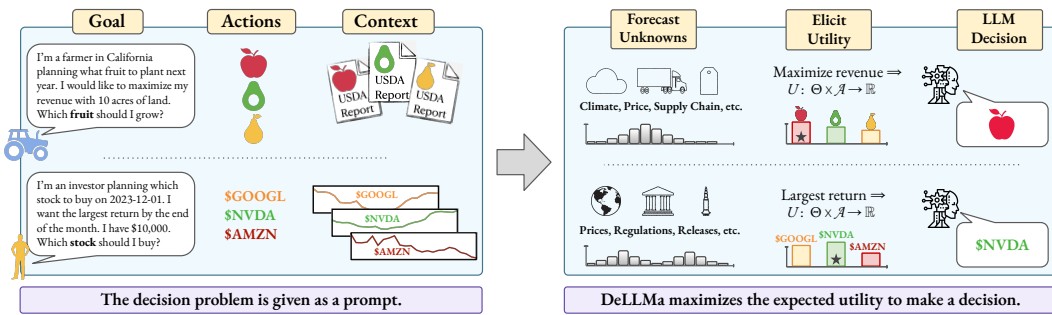

Figure 1: Given a decision problem and contextual information as a prompt, DeLLMa (*decision-making LLM assistant*) maximizes an expected utility to select an available action. We illustrate the key steps of DeLLMa on decision-making tasks in agriculture planning (*top*) and finance (*bottom*).

2024). These can be categorized into three main approaches: (1) instructing the model to generate intermediate reasoning traces (Wei et al., 2022; Yao et al., 2022; Zelikman et al., 2022; Zhuang et al., 2023; Ye et al., 2023; Shinn et al., 2024; Cheng et al., 2024), (2) decomposing a complex reasoning problem into tangible components (Zhou et al., 2022; Radhakrishnan et al., 2023; Yao et al., 2023; OpenAI, 2024), and most recently (3) scaling the number of parallel samples to be postprocessed into a final solution (Wang et al., 2022; Snell et al., 2024; Brown et al., 2024; OpenAI, 2024).

## 3 METHODS

**Preliminaries.** Suppose that a decision maker needs to make a choice between a set of options to achieve some goal—i.e., has a *decision problem*. We begin by formalizing such a decision problem, and afterwards describe how we approach decision making with LLMs. There are three main components to the decision problems that we will describe: *actions*, *states*, and *utilities*.

First, the *actions* are the possible options that a decision maker wishes to choose between. We use $\mathcal{A}$ to denote the space of actions, and $a \in \mathcal{A}$ for a single action. Second, the set of *unknown states* of nature are denoted $\Theta$. In our formulation, we define a state $\theta \in \Theta$ to be any latent variable whose true value is unknown, yet affects outcomes relevant to the decision maker's goals. To perform optimal decision making, one must act while accounting for uncertainty over these unknown states.

The third component involves the decision maker's preferences for different possible outcomes. We formalize our framework for decision making under uncertainty using *utility theory*, which can be viewed as "modeling the preferences of an agent as a real-valued function over uncertain outcomes" (Kochenderfer et al., 2022; Schoemaker, 1982; Fishburn, 1968). A key element is the *utility function* (in some formulations, this is instead given in terms of a *loss function L*). The utility function, denoted $U : \Theta \times \mathcal{A} \to \mathbb{R}$, assigns a scalar value to any state and action $(\theta, a) \in \Theta \times \mathcal{A}$. Intuitively, a higher utility means that the state-action pair yields a more-preferable outcome for the user.

The goal of the decision maker will be to choose a *final decision $a^* \in \mathcal{A}$*, which yields the highest possible utility, while accounting for uncertainty in the unknown states $\theta$.

**Decision Making with LLMs: Setup and Current Approaches.** We first describe the setting in which we intend our framework to operate. Suppose a human wishes to use this LLM assistant to help make a decision. They begin by describing a decision problem via a user prompt $\mathcal{P}$. We formalize a user prompt as a triplet $\mathcal{P} = (\mathcal{G}, \mathcal{A}, \mathcal{C})$, which includes: a natural language description of the user's goal $\mathcal{G}$, a list of $n$ actions $\mathcal{A} = (a_1, \ldots, a_n)$, and a passage of contextual information $\mathcal{C}$, which might be, *e.g.*, pages from a report, or a text-based representation of historical data.

Referring to the *agriculture planning* decision problem in Figure 1 as a running example, the goal $\mathcal{G}$ is for a farmer to maximize their revenue in the forthcoming year; the action set $\mathcal{A}$ lists the possible produce the farmer is considering planting (*e.g.*, apples, avocados, pears); and the context $\mathcal{C}$ consists of historical summaries of agricultural yields or information about the climate around the farm.

It is tempting to delegate such decision-making to LLMs with direct prompting. However, we observe that responses from conventional approaches, such as Self-Consistency and CoT, do not adequately balance available evidence, handle uncertain information, or align with user preferences; we show in Sec. 4 that these methods perform poorly, especially with increasing numbers of actions.

**DeLLMa: Decision-Making LLM Assistant.** To help encourage improved decisions under uncertainty, we propose a framework that guides an LLM to follow the scaffolding of classical decision theory. By restricting LLMs to this scaffold we can also explicitly see components of the decision-making process—*e.g.*, predictions of unknown states and utility function values—which provides human-auditability, allowing a user to identify why a given decision was made by the model.

In our initial formalization of this framework, we restrict ourselves to a slightly curtailed class of problems, and thus make a few simplifying assumptions. For example, we have assumed above that there are a discrete, enumerable set of $n$ possible actions, *i.e.*, $\mathcal{A} = (a_1, \ldots, a_n)$. We will also assume there is a discrete set of $m$ possible states, $\Theta = (\theta_1, \ldots, \theta_m)$, though $m$ may be quite large.

Our framework will use an LLM to produce a belief distribution over the unknown states, given the input context $\mathcal{C}$. We view this as a *posterior belief distribution* over the states, which we denote by $\pi(\theta \mid \mathcal{C})$. Implicitly, we are assuming that the LLM implies a *prior belief distribution* $\pi(\theta)$, given only the model weights or training data.

Our framework will also elicit a *utility function*, based in part on the description of the user's goals $\mathcal{G} \in \mathcal{P}$. This utility function assigns a scalar value to any state-action pair $(\theta, a)$. We denote this utility function as $U(\theta, a)$. Given these, the *expected utility* under our LLM of taking an action $a$, given some additional context $\mathcal{C}$, can be written

$$U_{\mathcal{C}}(a) = \mathbb{E}_{\pi(\theta|\mathcal{C})}[U(\theta, a)] = \sum_{\theta \in \Theta} \pi(\theta \mid \mathcal{C}) U(\theta, a). \tag{1}$$

Then, following the *expected utility principle* for rational decision making (Machina, 1987; Peterson, 2017), we select the *Bayes-optimal decision* $a^*$, which maximizes the expected utility, written

$$a^* = \arg\max_{a \in \mathcal{A}} U_c(a). \tag{2}$$

We call our framework **DeLLMa**, short for **De**cision-making **L**arge **L**anguage **M**odel **a**ssistant. DeLLMa carries out this sequence of four steps—state enumeration, state forecasting, utility elicitation, and expected utility maximization. A full description of DeLLMa is shown in the box below. In the following sections we give details on our specific implementation of each of these four steps.

---

**DELLMA: AN ASSISTANT FOR LLM DECISION MAKING UNDER UNCERTAINTY**

**Input**: Prompt $\mathcal{P} = (\mathcal{G}, \mathcal{A}, \mathcal{C})$ consisting of a user's goal $\mathcal{G}$, actions $\mathcal{A} = (a_1, \ldots, a_n)$, and context $\mathcal{C}$.

1. **State Enumeration**: Produce a list of $m$ *states* $\Theta = (\theta_1, \ldots, \theta_m)$, which are unknown quantities whose values are predicted to influence the user's goal $\mathcal{G}$. ▷ **Section 3.1**.

2. **State Forecasting**: For each state $\theta_j$, produce a probabilitic forecast $\pi(\theta_j \mid \mathcal{C})$, which describes the probability of different values of this state given context $\mathcal{C}$. ▷ **Section 3.2**.

3. **Utility Function Elicitation**: Produce a *utility function* $U : (\theta_j, a_i) \to \mathbb{R}$, which assigns a scalar value to each state-action pair $(\theta_j, a_i)$, based on the user's goal $\mathcal{G}$. ▷ **Section 3.3**.

4. **Expected Utility Maximization**: For each action $a_i$, compute the expected utility $U_{\mathcal{C}}(a_i) = \mathbb{E}_{\pi(\theta|\mathcal{C})}[U(\theta, a_i)]$, and return the decision $a^* = \arg\max_{i \in \{1, \ldots, n\}} U_{\mathcal{C}}(a_i)$. ▷ **Section 3.4**.

---

### 3.1 STATE ENUMERATION

As an initial demonstration of DeLLMa, our goal is to develop a simple implementation that performs well empirically; however, each component in the framework can be extended or made more sophisticated. We first describe the strategy that we adopt for enumerating a space of relevant latent states $\Theta = (\theta_1, \ldots, \theta_m)$. Given $\mathcal{P}$ as context, we prompt an LLM to identify $k$ *latent factors* that are predicted to influence the user's goal $\mathcal{G}$ (see §C.2 for details on this prompt). Each latent factor is a string (a word or phrase), which can be viewed as describing a dimension of our state space $\Theta$. We denote these $k$ latent factors as $(f_1, \ldots, f_k)$.

For each latent factor, we prompt our LLM to generate $\ell$ *plausible values* of the latent factor (empirically we find that it is sufficient to set $\ell$ to be small, e.g., $<5$). For a latent factor $f_j$, we denote its

plausible values as $\tilde{f}_j^{1:\ell}$, where each plausible value is also a string (a word or phrase). This process discretizes the state space, where each of the $k$ dimensions has $\ell$ bins. We find that this strategy, while simple, yields an empirically effective method for forecasting states (described in §3.2).

A single state $\theta_j$ in this state space consists of one plausible value from each of the $k$ latent factors, which we can denote by $\theta_j = \theta_j^{1:k} \in \Theta$. In total, this produces a discretized state space of size $|\Theta| = m = \ell^k$. While this state space is too large to enumerate explicitly, we develop a procedure to forecast probabilities for these states in a scalable manner.

## 3.2 STATE FORECASTING

In the next step of DeLLMa, we form a probabilistic forecast of the unknown states, given information contained in the context $\mathcal{C} \in \mathcal{P}$. A number of recent works have shown that LLMs are capable of returning well-calibrated forecasts with respect to some provided information (Halawi et al., 2024; Schoenegger et al., 2024) (see §2). Here, we develop a relatively simple forecasting method that we find works well empirically; though DeLLMa allows us to flexibly use other forecasting methods in the future (potentially leveraging search and retrieval of information). This step yields a distribution over the state space, which we can sample from to get a Monte Carlo estimate of the expected utility.

For each of the $k$ latent factors, and each of their $\ell$ possible values $\tilde{f}_j^{1:\ell}$, we prompt our LLM to assign a *verbalized probability score* $\in \{$ *very likely, likely, somewhat likely, somewhat unlikely, unlikely, very unlikely* $\}$. In total, we must assign $k \times \ell$ scores. We provide all prompts for this probability score procedure in Figure 11. We then define a dictionary $\mathcal{V}$ that maps each verbalized probability score to a numerical value. Similar strategies converting from verbalized to numeric scores have been used with success in prior work (Xiong et al., 2023; Tian et al., 2023). After normalization, this yields a distribution over the state space $\Theta$, assuming independence between the $k$ latent factors, which we posit for computational simplicity. We sample states from this distribution by iterating through each of the latent factors, sampling according to its approximate marginal probability, and concatenating the samples. In Sec. 4.1 we directly evaluate this state forecasting procedure to show that it yields well-calibrated forecasts in real-data scenarios, and we conduct an ablation study in Sec. 4.3.

The full procedure is shown in Algorithm 1. Here, the NORMALIZE function simply scales the weights instantiated in the marginal distribution to a well-defined probability mass function (PMF). We consider the sampled states to be from an LLM-defined proposal distribution $\pi^{\text{LLM}}(\theta \mid \mathcal{C})$, returned as output from Algorithm 1, which approximates the posterior belief distribution $\pi(\theta \mid \mathcal{C})$.

---

**Algorithm 1** STATEFORECAST

**Input:** LLM $\mathcal{M}$, user prompt $\mathcal{P} = (\mathcal{G}, \mathcal{A}, \mathcal{C})$, plausibilty score mapping $\mathcal{V}$, latent factors $\{f_1, \cdots, f_k\}$, and plausible values $\{\tilde{f}_1^{1:\ell}, \cdots, \tilde{f}_k^{1:\ell}\}$.

**for** $i = 1$ **to** $k$ **do**
  $\pi_i(\cdot \mid \mathcal{C}) \leftarrow \{\}$
  # Verbalized probability score
  $[v_1, \cdots v_\ell] \leftarrow \mathcal{M}(\mathcal{P}, f_i, \tilde{f}_i^{1:\ell})$
  **for** $j = 1$ **to** $\ell$ **do**
    $\pi_j(\tilde{f}_i^j \mid \mathcal{C}) \leftarrow \mathcal{V}[v_j]$
  **end for**
  $\pi_i(\cdot \mid \mathcal{C}) \leftarrow \text{Normalize}(\pi_i(\cdot \mid \mathcal{C}))$
**end for**
**return** $\pi^{\text{LLM}}(f_1, \cdots, f_k \mid \mathcal{C}) := \prod_{i=1}^k \pi_i(\cdot \mid \mathcal{C})$

---

**Algorithm 2** UTILITYELICITATION

**Input:** LLM $\mathcal{M}$, user prompt $\mathcal{P} = (\mathcal{G}, \mathcal{A}, \mathcal{C})$, proposal distribution $\pi^{\text{LLM}}(\theta \mid \mathcal{C})$, sample size $s$, minibatch size $b$, and overlap proportion $q$.
# Sample fixed states $\forall a \in \mathcal{A}$
$S_A \leftarrow \mathcal{A} \times \{\theta_i \mid \theta_i \sim \pi^{\text{LLM}}, 1 \leq i \leq \lfloor s/|\mathcal{A}| \rfloor\}$
$S_A \leftarrow \text{Shuffle}(S_A)$
$\Omega \leftarrow \{\}$  # Pairwise comparisons
**for** $i = 1$ **to** $s$ with step $\lfloor b \times (1-q) \rfloor$ **do**
  # Rank the minibatch
  $\mathcal{R} \leftarrow \mathcal{M}(\mathcal{P}, (\theta_i, a_i), \cdots, (\theta_{i+b}, a_{i+b}))$
  # Format into comparison
  $\Omega \leftarrow \Omega \cup \text{FormatRank}(\mathcal{R})$
**end for**
**return** $U(\cdot, \cdot) := \text{BradleyTerry}(\Omega) \in \mathbb{R}^s$

---

## 3.3 UTILITY FUNCTION ELICITATION

Next, we need a method to *elicit* (which is to say: *construct*) a utility function $U : \Theta \times \mathcal{A} \to \mathbb{R}$, which maps a state-action pair to a real value. An accurate utility function, which balances the preferences of a human user with respect to the goal that they describe, is difficult to define directly in a general-purpose manner. There is a long history of work on *utility elicitation* methods (Farquhar, 1984), which aim to construct a utility function from, e.g., pairwise preference data. Here, we combine these methods with large language models to try and automatically elicit a utility function.

We conduct the following procedure. We first sample states from the forecast state distribution $\pi^{\text{LLM}}(\theta \mid \mathcal{C})$, and from these form a set of state-action pairs. We group these pairs into minibatches,

and prompt our LLM to rank the elements of each minibatch, given the user's goal $\mathcal{G} \in \mathcal{P}$. This LLM-based ranking of items—where each item consists of an action and a particular instantiation of states—is a procedure that can be broadly applied, and LLMs have a history of being successfully used for similar comparisons (Lee et al., 2024; Qin et al., 2023). Based on these rankings, we are able to extract pairwise preferences, which we can use in classic utility elicitation algorithms.

We show this procedure in Algorithm 2 and discuss two implementations of the FormatRank step: Rank2Pairs and One-vs-All. Denoting $(\theta, a)_{(i)}$ as the $i$-th preferred state-action pair of the minibatch, Rank2Pairs converts a ranking of decreasing preference $\mathcal{R} = \big((\theta, a)_{(1)}, \cdots, (\theta, a)_{(b)}\big)$ to a list of pairwise comparisons by adding $(\theta, a)_{(i)} \succ (\theta, a)_{(j)}$ whenever $i < j$. In contrast, One-vs-All assumes that the LLM is indifferent towards all but the top-ranked state-action pair, *i.e.*, $\big\{(\theta, a)_{(1)} \succ (\theta, a)_{(i)} \mid \forall\, 2 \leq i \leq b\big\}$. This implementation may be desirable when accurate comparisons of certain suboptimal state-actions is challenging. We then make use of these preferences as training data for a Bradley-Terry model (Bradley & Terry, 1952) to elicit an approximate utility function $U : \Theta \times \mathcal{A} \to \mathbb{R}$ with respect to the sampled state-action pairs. Finally, we find two additional ingredients are beneficial for scaling our inference-time reasoning, which help improve the accuracy and computational efficiency of utility elicitation: *batching* and *variance reduction*.

**Batching.** We implement a batched inference procedure that slices state-action samples $S_A = \{(\theta, a)\}$ into overlapping minibatches for ranking. We ensure that $q\%$ of samples are shared between two consecutive minibatches drawn from $S_A$, where $q$ is a hyperparameter that modulates a minibatch's degree of exposure to the preference of the previous minibatch. Larger $q$ results in finer-grained preference at the cost of more queries. Effects of $q$ are ablated in Figure 3.

**Variance Reduction.** Directly sampling $|S_A|$ state values from the proposal distribution $\pi^{\text{LLM}}$ may lead to high-variance estimates of utilities. We instead sample $|S_A|/|\mathcal{A}|$ independent state values from $\pi^{\text{LLM}}$, create $|\mathcal{A}|$ duplicates, and pair them with each action $a$ (see Figure 5 in Appendix A).

In Section 4.3, we conduct ablation studies to validate our utility elicitation procedure, which shows high agreement rate between DeLLMa rankings and human preferences. We also study scaling laws of sample size and overlap percentage. We find that this type of specialized inference-time solution scales favorably against general-purpose systems such as OpenAI o1 (OpenAI, 2024).

### 3.4 EXPECTED UTILITY MAXIMIZATION

In the final step of DeLLMa, we compute the expected utility for each action, and then return the action that maximizes the expected utility. In particular, for each action, we compute a Monte Carlo estimate of the expected utility using state-action samples (drawn from the state forecast distribution $\pi^{\text{LLM}}(\theta \mid \mathcal{C})$), as well as the elicited utility function. Note that these calculations are all performed analytically (not via an LLM). We can then approximate the expected utility $U_{\mathcal{C}}(a)$ as

$$U_{\mathcal{C}}(a) = \mathbb{E}_{\pi(\theta|\mathcal{C})}\left[U(\theta, a)\right] \approx \frac{1}{|S|} \sum_{\theta \in S} U(\theta, a), \tag{3}$$

given a set of state samples $S \subseteq \Theta$ drawn from our LLM-defined state forecast distribution, which is an approximation of the LLM's posterior belief distribution about states given context $\mathcal{C}$, *i.e.*, $S \overset{i.i.d.}{\sim} \pi^{\text{LLM}}(\theta \mid \mathcal{C}) \approx \pi(\theta \mid \mathcal{C})$. After computing the expected utility $U_{\mathcal{C}}(a)$ for each action, DeLLMa returns the final decision: $a^* = \arg\max_{a \in \mathcal{A}} U_{\mathcal{C}}(a)$.

## 4 EXPERIMENTS

We evaluate the performance of DeLLMa on two decision making under uncertainty environments sourced from different domains: agricultural planning (**Agriculture**) and finance investing (**Stocks**). Both involve sizable degrees of uncertainty from diverse sources, and are representative of distinct data modalities (natural language and tabular) involved in decision making. We propose the following DeLLMa variants designed to assess algorithmic improvements, as outlined in Section 3:

- **DeLLMa-Pairs** is the method using all techniques in §3.3 and Rank2Pairs for utility elicitation.
- **DeLLMa-Top1** is identical to DeLLMa-Pairs, but replaces Rank2Pairs with One-vs-All.
- **DeLLMa-Naive** is a base version of DeLLMa-Pairs, where we sample multiple states per action and construct pairwise comparisons from a single batch (*i.e.*, no batching or variance reduction).

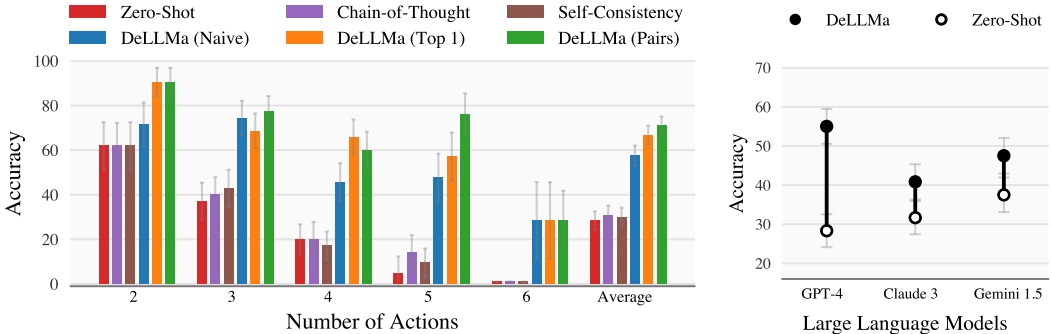

Figure 2: Results on the **Agriculture** environment. **Left:** DeLLMa variants outperform baseline methods for varying numbers of actions. **Right:** We see that DeLLMa yields a consistent improvement in decision-making accuracy across three families of leading LLMs.

For DeLLMa-Pairs and Top1, we allocate a *per action sample size* of 64 and a minibatch size of 32. We set the overlap proportion $q$ to 25% for the **Agriculture** dataset and 50% for the **Stocks** dataset due to budget constraints. For DeLLMa-Naive, we fix a *total sample size* of 50. As a default LLM in experiments (except for comparisons across different LLMs), we use GPT-4 (Achiam et al., 2023)[1].

We compare DeLLMa against three baselines—zero-shot, self-consistency, and Chain-of-Thought (where example prompts for these baselines are given in Figures 9, 10, 19, and 20):

- **Zero-Shot.** Only the goal $\mathcal{G}$, the action space $\mathcal{A}$, and the context $\mathcal{C}$ is provided. We adopt a greedy decoding process by setting temperature = 0.
- **Self-Consistency (SC)** (Wang et al., 2022). We use the same prompt as in zero-shot, but with temperature = 0.5 to generate a set of $K$ responses. We take the majority vote of the $K$ responses.
- **Chain-of-Thought (CoT)** (Wei et al., 2022). For decision-making tasks, there is no standard CoT pipeline. Inspired by workflows from decision theory, we create a prompting chain consisting of three steps: (1) ask for unknown factors that impact the decision; (2) given these, ask for their possibility of occurence; (3) then ask for a final decision. Such a mechanism is similar to the DeLLMa pipeline (see §3) but only consists of prompting.

**Evaluation Metrics.** For both datasets, our action spaces consist of a set of items, and we evaluate the performance of both DeLLMa and baseline methods by comparing the *accuracy* of their prediction from this set against the ground-truth optimal action (*i.e.*, the action that maximizes ground-truth utility). We also report *normalized utility*—*i.e.*, the ground-truth utility of the action chosen by a given method, normalized by the optimal ground-truth utility—in Appendix B. We defer more involved decision-making under uncertainty problems, such as constructing a weighted combination of actions (*i.e.*, a portfolio), to future works.

### 4.1 AGRICULTURE

**Data Acquisition.** We collect bi-annual reports published by the United States Department of Agriculture (USDA) that provide analysis of supply-and-demand conditions in the U.S. fruit markets[2]. To emulate real-life farming timelines, we use the report published in September 2021 as context for planning the forthcoming agricultural year. We additionally supplement these natural language contexts with USDA-issued price and yield statistics in California[3].

We define the utility of planting a fruit as its price × yield reported in the forthcoming year. We identify 7 fruits as our action set $\mathcal{A}$—apple, avocado, grape, grapefruit, lemon, peach, and pear—that are both studied in the September 2021 report, and endowed with these statistics in 2021 and 2022. We create decision making problems by enumerating all possible combinations of available fruits, resulting in 120 decision problem instances. For each decision-making instance, we use related sections of the USDA report and current-year price and yield statistics as context. See Appendix C.1 and Figure 8 for additional details on preprocessing, and Figure 13 for DeLLMa prompts.

---

[1]We use GPT-4 checkpoint `gpt4-1106-preview`.
[2]`www.ers.usda.gov/publications/pub-details/?pubid=107539`
[3]`www.nass.usda.gov/Quick_Stats`

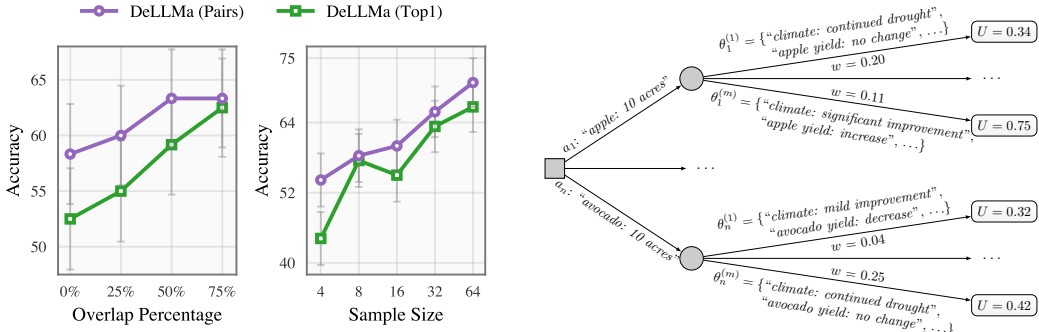

Figure 3: **Left:** Study on *sample size* and *overlap percentage* used by DeLLMa. Scaling the compute at test time produces better average accuracy. When ablating overlap percentage, we fix sample size at 16; when ablating sample size, we fix overlap percentage at 25%. **Right:** Illustration of the DeLLMa decision tree for the Agriculture dataset, showing two of the actions, and two of the sampled states per action. Each weight $w$ denotes the posterior probability $\pi(\theta_i^{(j)} \mid \mathcal{C})$.

**Main Result.** From Figure 2, we observe that all DeLLMa strategies consistently outperform baseline comparison methods, especially for larger action sets. We plot decision accuracy here, but include equivalent plots showing the utility of each strategy in Appendix B. Among DeLLMa variants, the performance of Pairs and Top1 are consistent across the board; both are significantly better than Naive. This observation shows the benefits of the algorithms outlined in §3.3. We give more detailed analyses in the ablation study below. We also show a comparison of DeLLMa deployed on multiple leading LLMs (GPT-4 (Achiam et al., 2023), Claude-3 (Anthropic, 2024), Gemini 1.5 (Reid et al., 2024)), with consistent performance improvements across multiple model families. We refer readers to Appendix C and supplementary material for details on data, prompts, and responses.

**How Good Are State Forecasts?** We first evaluate the quality of forecast distributions. We manually annotate a set of ground truth values for states that the LLMs forecast within DeLLMa, and then compare this with the forecast state distribution. We compare the average calibration (ECE), and the negative log-likelihood (NLL), two popular metrics for assessing the quality of a probabilistic forecast. As shown in Table 1, all models achieve reasonable calibration performances, which correlate with the overall DeLLMa prediction accuracy. We defer addi-

Table 1: An evaluation of state forecasting, showing ECE and NLL. Lower is better.

|  | ECE $\downarrow$ | NLL $\downarrow$ |
| --- | --- | --- |
| **GPT-4** | 0.062 | 1.11 |
| **Claude 3** | 0.142 | 1.20 |
| **Gemini 1.5** | 0.064 | 1.09 |

tional comparisons to Table 6, in which DeLLMa outperform randomized baselines. We conduct additional ablation studies to assess each DeLLMa module contributions in Section 4.3.

**Failure Modes of Baseline Methods.** One surprising observation is that, in the case of larger action sets, our baseline methods often *underperform* random guessing. A failure mode is that these methods elicit decisions that echo the sentiments presented in context, and they lack the ability to reason *what-if* scenarios that lead to utility changes. Our experiments on SC and CoT indicate that neither sampling reasoning paths, prompting the model to imagine the alternatives, nor augmenting an LLM with a posterior belief substantially enhance the model's ability to reason with *uncertainty*. By conditioning on sampled states, DeLLMa can avoid this pitfall while leveraging in-context learning to decide the preferred *state-action* pair. See Appendix C.4 and Figures 12, 17 and 18 for discussions on failure cases of baseline methods that are addressed by DeLLMa.

In addition to performance improvements, our structured approach is endowed with *human auditability*. In Figure 3 (right), we show an abbreviated decision network, with actions, states, sampled latent factors, and derived utilities constructed from outputs of a DeLLMa agent. This modular approach to decision making can facilitate transparency and trust of LLMs in high-stake scenarios.

## 4.2 STOCKS

Making decisions involving financial investing requires handling a variety of uncertainties; however, it is fundamentally different from the agriculture decision problems. Most evidently is the difference in input format—contexts for agriculture rely more on textual information (summarizations from

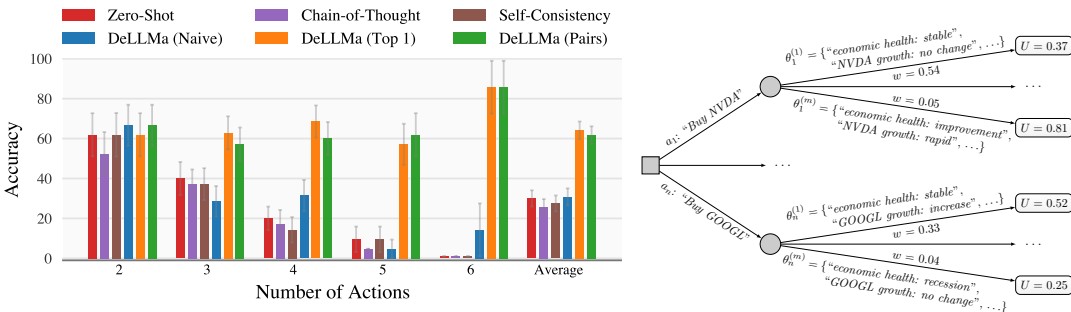

Figure 4: Results on the **Stocks** environment. **Left:** We see that on average, DeLLMa-Top1 outperforms all baselines. **Right:** Illustration of the DeLLMa decision tree for the Stocks dataset, showing two of the actions and two of the sampled states per action.

USDA reports) whereas contexts for stocks involve tabular data (historical prices per share). Stocks decision problems are well suited to test LLMs capability in reasoning on dynamic tabular data.

**Data Acquisition.** Similar to the setup in §4.1, the action space $\mathcal{A}$ consists of individual stocks and we curate a total of 120 decision problem instances of varying sizes. In our experiments, we choose popular stocks whose symbols are AMD, DIS, GME, GOOGL, META, NVDA and SPY. Unlike agriculture data where the context $\mathcal{C}$ are collected through USDA reports, we collect historical stock prices as the context for this problem. As illustrated in Figure 1, each stock is presented with 24 monthly price in history. In preventing possible data leakage and promoting LLMs to use their common-sense knowledge in making decisions, when using `gpt4-1106-preview` as the LLM checkpoint, historical price between December 2021 to November 2023 are provided as the context $\mathcal{C}$. These historical monthly prices are collected via Yahoo Finance[4] manually by the authors.

The goal of the LLM agent is to choose which stock to invest on December 1st, 2023, and sell on the last trading day of that month (December 29, 2023) so that the return is maximized. Detailed prompts are presented in Appendix D.1. We note that we only consider the simplistic setting—choosing *one* stock from a set of options $\{a_1, \cdots, a_n\}$.

**Main Results.** Similarly, we compare three variants of DeLLMa with the three popular baselines. Shown in Figure 4, most of the observations are consistent with those in the agriculture experiments—DeLLMa's variants outperform the baseline candidates. Unlike the agriculture setting, here DeLLMa-Naive only slightly improves over the baselines. We hypothesis this is due to the high volatility of stocks data, where inefficient sample size without variance reduction produces highly volatile predictors as well. This also validates the need for the design of the DeLLMa-Pairs and DeLLMa-Top1 methods.

Additionally, DeLLMa-Top1 performs better than DeLLMa-Pairs in the stocks data. This difference may stem from the simplicity of utility elicitation from only the top action choice, which requires less data processing and may mitigate noise compared to enumerating all state-action pairs. We hypothesize that in high-volatility data like stocks, LLMs may struggle with pairwise comparisons due to potential hallucination issues, particularly when the model attempts to rank options without a clear ground truth. By focusing on the top choice, DeLLMa-Top1 could avoid accumulating noise from these internal rankings, achieving better performance.

### 4.3 ABLATION STUDIES

**Scaling Laws for Inference-Time Compute.** Referring back to Figure 2, a potential explanation for the performance difference between DeLLMa-Naive and Pairs/Top1 is the discrepancy in sample size: for Naive, we allocate a fixed sample size (50) for all decision-making instances, and we allocate a fixed *per action sample size* (64) for Pairs and Top1. We eliminate this confounder in our ablation study, by noting that with *per action sample size* 8 (middle subfigure in Figure 3), DeLLMa-Pairs/Top1 achieve comparable performance to Naive, despite only receiving 16-48 samples in total for each problem instance.

---

[4] `finance.yahoo.com`

Table 2: Performance comparison across variations of our state forecasting procedure.

| | GPT-4 | Claude 3 | Gemini 1.5 |
|---|---|---|---|
| Uniform | 58.3% | 32.5% | 45.8% |
| Underspecified | 55.0% | 34.2% | 42.5% |
| Overspecified | 56.7% | 35.8% | 45.8% |
| DeLLMa | 60.0% | 40.8% | 47.5% |

Table 3: Performance comparison against SotA inference-time reasoning models.

| Task | DeLLMa ($n = 64$) | o1-preview |
|---|---|---|
| Agriculture | 73.3% | 33.3% |
| Stock | 64.2% | 35.0% |

Table 4: Results on a human evaluation of utility elicitation in DeLLMa. Details in §4.3.

| | GPT-4 | Claude 3 | Gemini 1.5 |
|---|---|---|---|
| Agreement % with Human | 68.4% | 65.3% | 65.7% |

In Figure 3 (left), we observe linear performance trends when scaling up overlap percentage and sample size. Intuitively, both higher overlap percentages (*i.e.*, more exposure between minibatches) and larger sample size lead to construction of finer-grained pairwise comparisons and thus high quality approximate utilities. Furthermore, DeLLMa-Pairs consistently outperforms Top1, implying that a nontrivial portion of the pairwise comparisons are meaningful. However, we note that the number of required API queries scales linearly with both parameters, and users are advised to choose these parameters that balance performance and cost. We report statistics for API queries and prompt lengths for our methods in Table 5 of Appendix B.

**State Forecasting Ablation.** We now assess the quality of the state forecasting procedure by evaluating DeLLMa in the presence of low-quality or extraneous state information. On the agriculture dataset, we report in Table 2 the ablation results from 3 variants for the propoal distribution $\pi^{\text{LLM}}$: uniform, underspecified, and overspecified. We fix a *per action sample size* of 16 and an *overlap percentage* of 25%, and refer readers to Appendix E for additional details.

We observe that the performance of the modified forecasting methods in GPT-4 and Gemini 1.5 are similar to full DeLLMa performance, and are better than our baseline methods. With these models, DeLLMa appears to be robust against misspecified, insufficiently representative, and extraneous forecast states. However, with Claude 3, ablation performance is more akin to zero-shot baselines. This discrepancy may be due to Claude's weaker performance in DeLLMa (c.f. Figure 2). Overall, our state forecasting procedure appears to be beneficial for DeLLMa performance.

**Evaluation Against OpenAI o1.** In Section 4, we have compared different prompting approaches while fixing the LLM backbone. We now evaluate OpenAI o1—a class of more advanced models adept in inference-time reasoning—against DeLLMa. Table 3 shows that our best performances (attained with a *per action sample size* of 64 and *overlap percentage* of 25%) can outperform o1 with the *zero-shot* prompt (c.f. Figures 9 and 22) by a wide margin, while attaining similar costs (DeLLMa: $0.09 to $0.37 vs. o1: $0.21 per instance)[5]. This study indicates the benefits of specialized inference-time reasoning for decision making under uncertainty.

**Human Evaluation for Preference Ranking.** Lastly, we perform a study to evaluate the quality of utility elicitation in DeLLMa. We curate a dataset of 412 pairwise state-action samples; human annotators (the paper authors and 5 external volunteers) are then shown pairs of these state-action samples and asked to annotate a preference, based on the decision prompt $\mathcal{P}$. These pairs are presented in *shuffled* order to eliminate potential positional bias. We then compute the agreement rate between the pairwise annotations given by DeLLMa and those given by the annotators. Although this step is noisy, we see a strong agreement between LLM and human annotations, as shown in Table 4, across multiple LLMs. We also find that our inter-annotator agreement rate is 67.0% ($\pm 6.3\%$), which is on par with the human-LLM agreements. Additional details of our annotation procedure are reported in Appendix F.

## 5 CONCLUSION

We propose DeLLMa, a framework designed to harness LLMs for decision making under uncertainty in high-stakes settings. This is a structured approach in which we provide a process for inference-time reasoning in LLMs that follows the principles of classical decision theory. We then develop a feasible implementation with LLMs. Through experiments on real datasets, we highlight a systematic failure of popular prompting strategies when applied to this type of decision making task, and demonstrate the benefits of our approach to address these issues. The modularity of our framework avails many possibilities, most notably auditability and using LLMs for a broader spectrum of probablistic reasoning tasks.

---

[5]DeLLMa costs can be further reduced by 50% with the OpenAI batch API.

## REPRODUCIBILITY STATEMENT

To facilitate reproducibility, we have included the following artifacts within our paper and supplementary materials:

**Data.** All DeLLMa data, prompts, and LLM inference results for main experiments are given in Section 4. We also include all prompts in Appendices C.2 and D.1.

**Code.** Our implementations for the zero-shot, self consistency, Chain-of-Thought, and DeLLMa methods are included in the supplementary material. While these inference results are included, reproducing them require OpenAI, Anthropic, and Gemini API access. We provide evaluation script in `results.sh` for reproducing all numerical results.

## ACKNOWLEDGEMENT

We thank Yu Feng, the USC NLP Group, and anonymous reviewers for valuable insights, discussions, and comments.

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

# APPENDIX

## A    UTILITY ELICITATION DETAILS

In Figure 5, we show an illustration of the overlapped batching and variance reduction strategies that DeLLMa uses in its utility elicitation procedure (described in detail in Section 3.3).

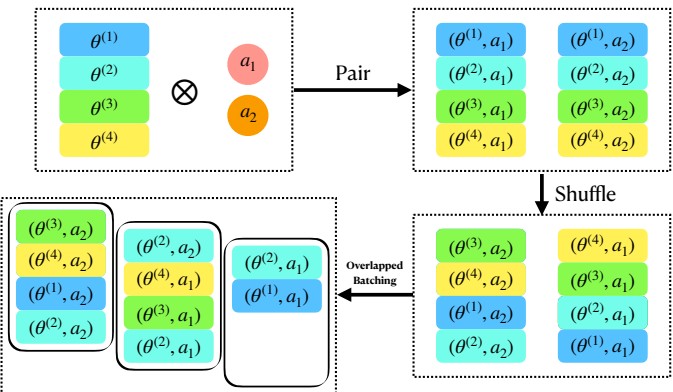

Figure 5: Schematic diagram of overlapped batching and variance reduction.

## B    ADDITIONAL RESULTS

In Figures 6 and 7 we show the normalized utility (*i.e.*, ground-truth utility of the chosen action, normalized by the maximum ground-truth utility) of each method. These results can be contrasted against the accuracy of each method's prediction of the optimal action in Figures 2 and 4.

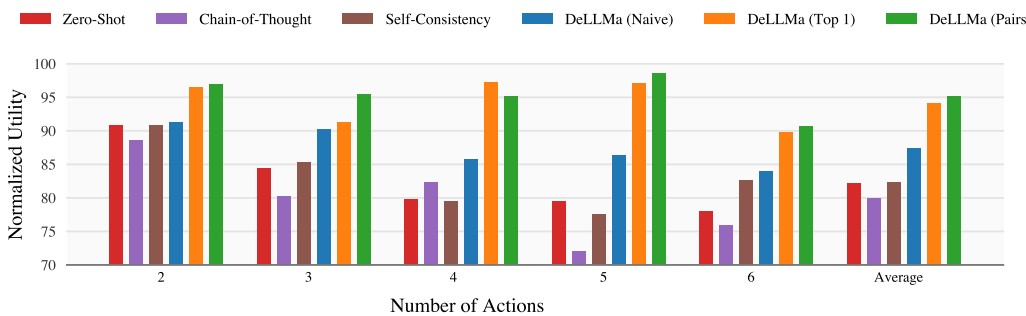

Figure 6: Normalized utility for the **Agriculture** dataset. Across all action sizes, DeLLMa-Pairs/Top1 outperforms other methods, including DeLLMa-Naive. Similar to accuracy, DeLLMa-Pairs slightly outperforms Top1, which is consistent with our hypothesis that the complete ranking generated from GPT-4 is meaningful.

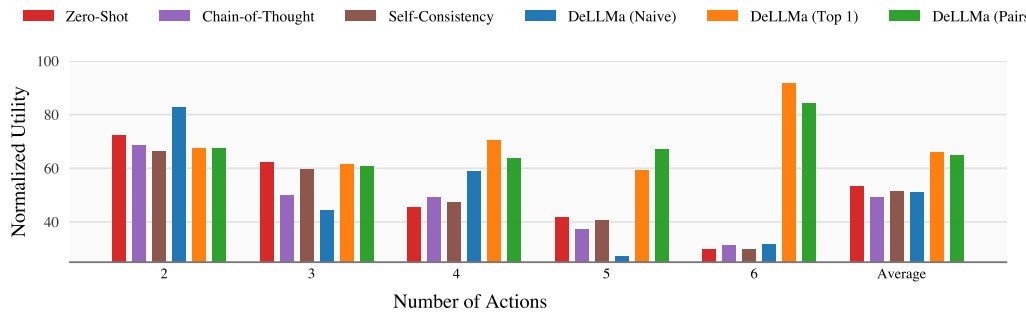

Figure 7: Normalized utility for the **Stock** dataset. DeLLMa-Top1 is competitive againt Pairs, suggesting that financial decision making may be a challenging scenario for GPT-4 to elicit preference unequivocally.

Aside from performance improvements, we also compare the cost measured in terms of prompt length and API calls in Table 5. For the **Agriculture** dataset, DeLLMa-Naive is comparable to SC in terms of prompt length and significantly outperform the latter, while only requiring 20% of API calls. DeLLMa-Pairs/Top1 can push the performance further, but incurs a higher cost, especially for large sample sizes.

Table 5: Number of GPT-4 API calls and word counts per decision-making instance (with action set size 4 for the **Agriculture** dataset) across all methods discussed in Section 4.1. For DeLLMa-Naive, we set the total sample size to 50. For DeLLMa-Pairs, we fix the overlap percentage to 25% and vary the *per action sample size* from 16 to 64. DeLLMa-Top1 has the same statistics as DeLLMa-Pairs since they only differ in post processing.

|  | **Zero-Shot** | **CoT** | **SC** | **D-Naive** | **D-Pairs (16)** | **D-Pairs (64)** |
|---|---|---|---|---|---|---|
| **API Calls** | 1 | 3 | 5 | 1 | 3 | 10 |
| **Token Counts** | 495 | 1946 | 2477 | 2629 | 5181 | 20639 |

We provide additional results on forecasting performance measured by the expected calibration error (ECE) to supplement Table 1, by focusing on two sets of statistics in Table 6:

- ECE under the uniform distribution forecast (ECE-Uniform).
- ECE under the LLM forecast, but with random choice as the ground truth (ECE-Random). We report the standard deviation from 100 random trials.

To calculate the ECE, for a fixed set of latent factors (e.g., fruit price change, climate conditions), we manually find and annotate ground truth values — using the USDA report and web search. We can then compute metrics such as the expected calibration error (ECE) for forecasts of these latent factors given by a model. Together, these results indicate that our state forecasting algorithm attains much improved performance in comparison with these baselines.

Table 6: Comparison of ECE across different proposal distributions against randomized baselines.

|  | **ECE** | **ECE-Random (±SD)** | **ECE-Uniform** |
|---|---|---|---|
| **DeLLMa (GPT-4)** | 0.062 | 0.135 ± 0.092 | 0.333 |
| **DeLLMa (Claude 3)** | 0.142 | 0.157 ± 0.084 | 0.333 |
| **DeLLMa (Gemini 1.5)** | 0.064 | 0.113 ± 0.083 | 0.333 |

# C  ADDITIONAL DETAILS FOR AGRICULTURE

## C.1  DATASET CURATION

To reduce context length, we first extract executive and per-fruit summaries of the report, reducing the context length from 8,721 words to around $< 700$ words. These summaries are constructed on

a *per-model* basis: each LLM (GPT-4, Claude-3, and Gemini 1.5) is prompted to generate its own summaries. We proof-read these summaries to ensure that they do not contain any factual errors. For each decision making instance, we use the executive summary, the summaries relevant to the fruits in consideration, and their current-year price and yield statistics as context. We provide these summaries in our supplementary material, and refer readers to the prompt for summarizing the report in Appendix C.2, and report concrete summaries and statistics in supplementary materials.

## C.2 PROMPT USED BY AGRICULTURE

**Summary Prompt** In Figure 8 we present the prompt used for summarizing the context in our **Agriculture** experiments. This prompt is shared across all models tested.

**Zero-Shot Prompt** In Figure 9 we present an example prompt for zero-shot experiments consisting of $\mathcal{P} = (\mathcal{G}, \mathcal{C}, \mathcal{A})$, with GPT-4 summaries generated from Figure 8 as context. We fix the prompt format for all LLMs and select the corresponding summaries for each LLM.

**Self-Consistency Prompt** We adopt the same zero-shot prompt as the one shown in Figure 9, but take a majority vote from $K$ reasoning paths as the final prediction. Our preliminary study finds that LLMs tend to be very confident in their decisions, and even with much higher temperature than 0.5, often all $K$ independent runs yield consistent decisions. We thus set $K = 5$ to balance cost and performance.

**Chain-of-Thought Prompt** We design a multi-prompt CoT procedure that closely emulates our DeLLMa agent, consisting of three steps: (1) ask for unknown factors that impact the decision; (2) given these, ask for their possibility of occurence; (3) then ask for a final decision. This procedure differs from DeLLMa agents as it does not use an external program for *utility elicitation* and *expected utility maximization*, but delegates each step of the process to an LLM. Example prompt can be found in Figure 10.

**DeLLMa Prompt** Similar to the CoT Prompt, DeLLMa is a multi-prompt procedure (1) ask for unknown factors that impact the decision; (2) given these, ask for their possibility of occurence (*i.e. a belief distribution*). Instead of directly deciding on an action, DeLLMa samples *state-action* pairs (via an external program) from this belief distribution, and leverage an LLM to elicit a utility function $U : \mathcal{S} \times \mathcal{A} \to \mathbb{R}$ from pairwise comparisons. From here, we use another external program to search for an action that maximizes the expected utility.

In Figures 8 and 11 we present the prompt for *state enumeration* and *state forecasting* (*i.e.* step (1) and (2)) for the **Agriculture** dataset. Our implementation combines these two modules, but in the future they should be separate as our framework continues to progress. For example, *state enumeration* may leverage retrieval-augmented generation Lewis et al. (2020) for generating high-quality unknown factors, and *state forecasting* may resort to tool usage to delibrate calibrated belief distributions from quanitative analysis.

In Figure 13, we provide a DeLLMa prompt to generate ranked comparisons from sampled *state-action* pairs. These state action pairs are sampled from the belief distribution via an external program, written in Python. We provide all our implementations in the supplementary material.

## C.3 AGRICULTURE DATASET RESPONSES

**Summaries** See Figure 14 for formatted GPT-4 summaries from the USDA agricultural report and statistics. Claude-3 and Gemini summaries follow the exact same format, but differ in LLM generated contents.

**Zero-Shot Responses** See Figures 12, 17 and 18 for sample GPT-4 responses to zero-shot prompts.

**Self-Consistency Responses** SC responses follow the exact same format as those in Figures 12, 17 and 18.

**Chain-of-Thought Responses**   See Figure 15 for a curtailed version of GPT-4 CoT response.

**DeLLMa Responses**   See Figure 16 for a curtailed version of GPT-4 DeLLMa response.

### C.4   FAILURE CASES ON THE AGRICULTURE DATASET

In Section 4.1, we postulate that a failure mode of our baseline approaches is that they lack the ability to perform probablistic reasoning, but are rather following the sentiment presented in context. Here, we qualitatively test this hypothesis, by showcasing prompts and responses from our Zero-Shot experiments. We present instances in Figures 12, 17 and 18 that are not able to make optimal prediction with baseline methods, but are solved by DeLLMa agents.

Within each figure, we first present $\mathcal{P}$, followed by the (incorrect) decision and explanation generated from the Zero-Shot baseline, and then present the top-ranked state action pair generated by DeLLMa along with its explanation.

In the case of deciding between apple and grapefruit in Figure 12, GPT-4 presumes that a high price for grapefruit is sustainable due to the presence of extreme weather conditions in the *previous year*. With CoT, the model reasons with some counterfactual scenarios, such as more suitable weather for the forthcoming year, but not in a systematic way. With DeLLMa, we provide these a list of counterfactual scenarios as context, and leverage the model's ability for situational thinking and ranking to elicit an informative preference. We make similar observations for Figures 17 and 18.

## D   ADDITIONAL DETAILS FOR STOCKS

### D.1   PROMPT USED BY STOCKS

**Zero-Shot Prompt**   Similar to the agriculture setup, we first present a sample zero-shot prompt in Figure 19. The zero-shot prompt consists of three main parts (see Figure 1 for visual illustrations): (1) an enumeration of action spaces (here AMD or GME), (2) a provided context (here, historical stock prices between December 2021 to November 2023), and (3) the goal of the user (choose only one stock to maximize their profit via investing in stocks).

**Self-Consistency Prompt**   SC prompts are identical to zero-shot prompts.

**Chain-of-Thought Prompt**   Similarly, we examplify a prompt for CoT. Notably, we break the chain into three parts in Figure 20: (1) Enumerating unknown factors, (2) Enumerating $\pi^{\mathrm{LLM}}(\cdot \mid \mathcal{C})$, and (3) making the final decision given the previous responses and context.

**DeLLMa Prompt**   Finally, we showcase the DeLLMa ranking prompts where the LLM is asked to provide comprehensive rankings of given state-action pairs.

### D.2   STOCK DATASET RESPONSES

**Zero-Shot Responses**   See Figure 22 for sample GPT-4 responses to zero-shot prompts.

**Self-Consistency Responses**   SC responses follow the exact same format as those in Figure 22.

**Chain-of-Thought Responses**   See Figure 23 for a curtailed version of GPT-4 CoT response.

**DeLLMa Responses**   See Figure 24 for a curtailed version of GPT-4 DeLLMa response.

## E   ADDITIONAL DETAILS FOR ABLATION STUDIES

- Uniform: we maintain the same set of states, but replace the forecast distribution $\pi^{\mathrm{LLM}}$ with a uniform distribution.

- Underspecified: we reduce the diversity of our set of states by removing a latent factor $f_k$, chosen at random, and then perform state forecasting as usual.
- Overspecified: in addition to the original set of latent factors, we add a small set of additional factors, which are intentionally unrepresentative or redundant. This represents a case where the latent factors contain extraneous information, which may confuse the LLM backbones.

## F  DETAILS ON ANNOTATION PIPELINE

For each pair of state-action values, we ask human evaluators (paper authors and 5 external volunteers) to annotate which state-action tuple is more preferred to the other, and compare our annotations against preferences elicited by LLMs. To reduce bias in this procedure, we present these pairs in *shuffled* orders, *i.e.* it is not always true that $(s,a)_1 \succ (s,a)_2$ if $\{(s,a)_1, (s,a)_2\}$ is presented to the annotators. For each pair, we ask the annotators to either label them as 1 (state-action tuple 1 is preferred), 2 (state-action tuple 2 is preferred), or 0 (uncertain). We then evaluate annotator-LLM agreement after evaluation is concluded. In total, this procedure yields 412 annotations.

From these annotation results, we measure the agreement between each pair of annotators, and compute inter-annotator agreement by micro-averaging all agreements. This procedure yields an average agreement of 67.0% ($\pm 6.3\%$), which is on par with human-LLM agreements.

## G  LICENSE FOR EXISTING ASSETS

**Large Language Models**   DeLLMa is built on top of frontier class LLMs such as GPT-4 (Achiam et al., 2023), Claude-3 (Anthropic, 2024), and Gemini 1.5 (Reid et al., 2024). We are not aware of their licenses, but ascertain that we abide to their respective terms of service (ToS)[6] for our usage.

**Agriculture Dataset**   The **Agriculture** dataset is curated from statistics and reports published by the U.S. Department of Agriculture. These assets are freely accessible under the Freedom of Information Act. We ascertain that we abide to the USDA ToS[7].

**Stock Dataset**   The **Stock** dataset is curated with the Yahoo Finance API[8]. We ascertain that we abide to the Yahoo Developer API ToS[9].

## H  SOCIETAL IMPACT AND LIMITATIONS

Rational decision making under uncertainty has been extensively studied across many disciplines, but remains elusive even for humans. Our work bears significant societal consequences as we live in a world where humans increasingly rely on intelligent assistants for diverse problems. Unfortunately, existing foundation models and LLMs are not capable of acting optimally under uncertainties yet, which may induce harm if the general public delegate these models for decision making. These risks are exacerbated when intelligent assistants operate as a black box without adequate transparency. DeLLMa serves as a first step towards an approach that builds trust for humans to rely on these systems to make important decisions under uncertainty.

While our approach sensibly improves the performance of LLMs for decision making under uncertainty, deploying a prototype system in the wild without additional guardrail could incur significant financial losses, in case of a suboptimal decision. We expect users to conduct extensive field tests for the feasibility of such systems, or leverage a hybrid approach that integrates analyst judgements for optimal decision making.

Another limitation of our approach stems from constrained state and action spaces. Our framework currently operates on a small number of discrete state and action spaces, due to the context window

---

[6]GPT-4 ToS, Claude-3 ToS, and Gemini 1.5 ToS.
[7]https://www.usda.gov/policies-and-links
[8]finance.yahoo.com
[9]Yahoo Finance API ToS

size of state-of-the-art LLMs. This drawback may potentially limit the applicability of our system in more complex use cases, or scenarios that require sequential decision making.

Finally, our current experiments focus on specific domains (agriculture and stocks) as representative, controlled environments for testing decision-making under uncertainty. Future work will involve evaluating DeLLMa across a broader set of domains to better understand its strengths and limitations in diverse, real-world applications.

---

EXAMPLE **SUMMARIZATION** PROMPT FOR AGRICULTURE

```
Below is an agriculture report published by the USDA:
<USDA Fruits & Nuts Report, omitted for brevity>[a]
Please write a detailed summary of the report. You should format your response as a JSON object.
The JSON object should contain the following keys:
- 'summary': a string that summarize, in detail, the overview of the report. Your summary should
include price, yield, production, and other information relevant to a farmer making decisions about
what to plant. You should also include key factors, such as weather, supply chain, and demand, that
affect the market.
- 'apple': a string that describes, in detail, information pertaining to apple in the report. You
should include information on apple prices and production, as well as factors that affect them.
- 'avocado': a string that describes, in detail, information pertaining to avocado in the report.
You should include information on avocado prices and production, as well as factors that affect them.
- 'grape': a string that describes, in detail, information pertaining to grape in the report. You
should include information on grape prices and production, as well as factors that affect them.
- 'grapefruit': a string that describes, in detail, information pertaining to grapefruit in the
report. You should include information on grapefruit prices and production, as well as factors that
affect them.
- 'lemon': a string that describes, in detail, information pertaining to lemon in the report. You
should include information on lemon prices and production, as well as factors that affect them.
- 'peach': a string that describes, in detail, information pertaining to peach in the report. You
should include information on peach prices and production, as well as factors that affect them.
- 'pear': a string that describes, in detail, information pertaining to pear in the report. You
should include information on pear prices and production, as well as factors that affect them.

- 'factors': a list of strings that enumerates the factors that affect the market, based on the

report. You should include at least 5 factors, ranked in decreasing order of importance.
```

Figure 8: EXAMPLE **SUMMARIZATION** PROMPT FOR AGRICULTURE

---

[a]https://www.ers.usda.gov/publications/pub-details/?pubid=107539

EXAMPLE **ZERO-SHOT** PROMPT FOR AGRICULTURE

Below is an agriculture report published by the USDA. It gives an overview of the fruit and nut market in the United States, with an additional focus on information pertaining to apple, avocado. Market Overview:  the USDA report indicates a general increase in U.S. production of major noncitrus fruits for 2021, with apples, grapes, peaches, cranberries, and sweet and tart cherries seeing a rise in production, while pear production is forecasted to decline.  the impact of extreme weather events and california's ongoing drought on crop yields is uncertain.  fruit and tree nut grower price indices remain high, with fluctuations throughout 2021.  the consumer price index for fresh fruit also increased, suggesting higher retail prices.  the northwest heat dome has introduced production uncertainty, particularly for tree fruits.  the U.S. citrus season ended with declines in all commodities except california tangerines, and citrus prices are higher.  tree nut supplies are forecasted to be down from the previous year's record, with smaller almond and walnut crops expected to increase grower prices.  factors such as weather conditions, supply chain issues, and demand are influencing the market.
- Apple:
    - Product Summary:  apple production is forecasted to be up 3 percent from 2020/21 but down 5 percent from 2019/20.  washington state's crop is expected to be larger, but there is concern over heat damage.  export markets may remain sluggish due to high tariffs and shipping challenges, potentially pushing more apples into the domestic market and lowering prices.  processing prices may rise due to declines in new york and michigan, which account for a significant portion of processed apples.
    - California Price and Yield Statistics:  the average apple yield is 19,000 LB / ACRE and the average price per unit is 0.244 $ / LB.
- Avocado:
    - Product Summary:  california avocado production has decreased, with wildfires and water restrictions impacting yields.  however, U.S. avocado consumption has increased significantly, with imports from mexico and peru growing substantially.  mexico dominates the U.S. avocado market, with imports peaking from may through july.  peruvian imports compete during the summer months, traditionally a period of lower mexican imports.
    - California Price and Yield Statistics:  the average avocado yield is 2.87 TONS / ACRE and the average price per unit is 2,430 $ / TON.
I'm a farmer in California planning what fruit to plant next year.  I would like to maximize my profit with '10' acres of land.
Below are the actions I can take:
    Action 1.  apple:  10 acres
    Action 2.  avocado:  10 acres

I would like to know which action I should take based on the information provided above.

Figure 9: EXAMPLE **ZERO-SHOT** PROMPT FOR AGRICULTURE

---

EXAMPLE **CHAIN-OF-THOUGHT** PROMPT FOR AGRICULTURE

**PROMPT 1**
Below is an agriculture report published by the USDA. It gives an overview of the fruit and nut market in the United States, with an additional focus on information pertaining to apple, avocado. Market Overview: the usda report indicates a general increase in u.s. production of major noncitrus fruits for 2021, with apples, grapes, peaches, cranberries, and sweet and tart cherries seeing a rise in production, while pear production is forecasted to decline. the impact of extreme weather events and california's ongoing drought on crop yields is uncertain. fruit and tree nut grower price indices remain high, with fluctuations throughout 2021. the consumer price index for fresh fruit also increased, suggesting higher retail prices. the northwest heat dome has introduced production uncertainty, particularly for tree fruits. the u.s. citrus season ended with declines in all commodities except california tangerines, and citrus prices are higher. tree nut supplies are forecasted to be down from the previous year's record, with smaller almond and walnut crops expected to increase grower prices. factors such as weather conditions, supply chain issues, and demand are influencing the market.
– Apple:
    – Product Summary: apple production is forecasted to be up 3 percent from 2020/21 but down 5 percent from 2019/20. washington state's crop is expected to be larger, but there is concern over heat damage. export markets may remain sluggish due to high tariffs and shipping challenges, potentially pushing more apples into the domestic market and lowering prices. processing prices may rise due to declines in new york and michigan, which account for a significant portion of processed apples.
    – California Price and Yield Statistics: the average apple yield is 19,000 LB / ACRE and the average price per unit is 0.244 $ / LB.
– Avocado:
    – Product Summary: california avocado production has decreased, with wildfires and water restrictions impacting yields. however, u.s. avocado consumption has increased significantly, with imports from mexico and peru growing substantially. mexico dominates the u.s. avocado market, with imports peaking from may through july. peruvian imports compete during the summer months, traditionally a period of lower mexican imports.
    – California Price and Yield Statistics: the average avocado yield is 2.87 TONS / ACRE and the average price per unit is 2,430 $ / TON.
I'm a farmer in California planning what fruit to plant next year. I would like to maximize my profit with '10' acres of land.
Below are the actions I can take:
Action 1. apple: 10 acres
Action 2. avocado: 10 acres
First think about the unknown factors that would affect your final decisions.

**PROMPT 2**
<SAME CONTEXT>
Now I have enumerated the unknown factors that would affect my final decisions:
<RESPONSE FROM PROMPT 1 CONTAINING LLM ENUMERATED UNKOWN FACTORS>
**Given these unknow factors, think about the possiblity that each factor would occur within a month.**

**PROMPT 3**
<SAME CONTEXT>
Now I have enumerated the unknown factors that would affect my final decisions:
<RESPONSE FROM PROMPT 1 CONTAINING LLM ENUMERATED UNKOWN FACTORS>
I also empirically estimated the possibility of occurrence of each possible factor:
<RESPONSE FROM PROMPT 2 CONTAINING LLM BELIEF DISTRIBUTION OVER UNKOWN FACTORS>
**Given these unknow factors and the possibility estimates of these factors' occurrences, think about your final decision.**

**I would like to know which action I should take based on the information provided above.**

Figure 10: EXAMPLE **CHAIN-OF-THOUGHT** PROMPT FOR AGRICULTURE

---

**EXAMPLE BELIEF STATE ELICIATION PROMPT FOR AGRICULTURE**

Below is an agriculture report published by the USDA. It gives an overview of the fruit and nut market in the United States, with an additional focus on information pertaining to apple, avocado, grape, grapefruit, lemon, peach, pear.
Market Overview: the usda report indicates a general increase in u.s. production of major noncitrus fruits for 2021, with apples, grapes, peaches, cranberries, and sweet and tart cherries seeing a rise in production, while pear production is forecasted to decline. the impact of extreme weather events and california's ongoing drought on crop yields is uncertain. fruit and tree nut grower price indices remain high, with fluctuations throughout 2021. the consumer price index for fresh fruit also increased, suggesting higher retail prices. the northwest heat dome has introduced production uncertainty, particularly for tree fruits. the u.s. citrus season ended with declines in all commodities except california tangerines, and citrus prices are higher. tree nut supplies are forecasted to be down from the previous year's record, with smaller almond and walnut crops expected to increase grower prices. factors such as weather conditions, supply chain issues, and demand are influencing the market.
- Apple:
  - Product Summary: apple production is forecasted to be up 3 percent from 2020/21 but down 5 percent from 2019/20. washington state's crop is expected to be larger, but there is concern over heat damage. export markets may remain sluggish due to high tariffs and shipping challenges, potentially pushing more apples into the domestic market and lowering prices. processing prices may rise due to declines in new york and michigan, which account for a significant portion of processed apples.
  - California Price and Yield Statistics: the average apple yield is 19,000 LB / ACRE and the average price per unit is 0.244 $ / LB.
- avocado:
  - Product Summary: california avocado production has decreased, with wildfires and water restrictions impacting yields. however, u.s. avocado consumption has increased significantly, with imports from mexico and peru growing substantially. mexico dominates the u.s. avocado market, with imports peaking from may through july. peruvian imports compete during the summer months, traditionally a period of lower mexican imports.
  - California Price and Yield Statistics: the average avocado yield is 2.87 TONS / ACRE and the average price per unit is 2,430 $ / TON.
...
- pear:
  - Product Summary: pear production is forecasted to be similar to the previous year, with losses in washington but gains in oregon and california. the impact of the northwest heat wave on production levels is still uncertain, but traditional pear trees with large canopies may offer some protection from heat damage.
  - California Price and Yield Statistics: the average pear yield is 15.6 TONS / ACRE and the average price per unit is 565 $ / TON.
I would like to adopt a decision making under uncertainty framework to make my decision. The goal of you, the decision maker, is to choose an optimal action, while accounting for uncertainty in the unknown state. The first step of this procedure is for you to produce a belief distribution over the future state. The state is a vector of 16 elements, each of which is a random variable. The state variables are enumerated below:
  - climate condition: the climate condition of the next agricultural season in California
  - supply chain disruptions: the supply chain disruptions of the next agricultural season in California
  - apple price change: the change in price per unit of apple for the next agricultural season in California
  - apple yield change: the change in yield of apple for the next agricultural season in California
  - avocado price change: the change in price per unit of avocado for the next agricultural season in California
  - avocado yield change: the change in yield of avocado for the next agricultural season in California
    ...
  - pear yield change: the change in yield of pear for the next agricultural season in California
You should format your response as a JSON object with 16 keys, wherein each key should be a state variable from the list above.
Each key should map to a JSON object with 3 keys, each of which is a string that describes the value of the state variable. Together, these keys should enumerate the top 3 most likely values of the state variable. Each key should map to your belief verbalized in natural language. If the state variable is continuous (e.g. changes to a quantity), you should discretize it into 3 bins.
You should strictly choose your belief from the following list: 'very likely', 'likely', 'somewhat likely', 'somewhat unlikely', 'unlikely', 'very unlikely'. For example, if one of the state variable is 'climate condition', and the top 3 most likely values are 'drought', 'heavy precipitation', and 'snowstorm', then your response should be formatted as follows:

```
{
    "climate condition": {
        "drought": "somewhat likely",
        "heavy precipitation": "very likely",
        "snowstorm": "unlikely"
    },
```

Figure 11: EXAMPLE BELIEF STATE ELICIATION PROMPT FOR AGRICULTURE

}

---

---

EXAMPLE **FAILURE CASE** FOR AGRICULTURE

**Ground Truth: Apple**
**Prompt** (Market overview omitted for brevity)
- apple:
    - Product Summary:  apple production is forecasted to be up 3 percent from 2020/21 but down
5 percent from 2019/20.  washington state's crop is expected to be larger, but there is concern
over heat damage.  export markets may remain sluggish due to high tariffs and shipping challenges,
potentially pushing more apples into the domestic market and lowering prices.  processing prices may
rise due to declines in new york and michigan, which account for a significant portion of processed
apples.
    - California Price and Yield Statistics:  the average apple yield is 19,000 LB / ACRE and the
average price per unit is 0.244 $ / LB.
- grapefruit:
    - Product Summary:  grapefruit production has been significantly affected by winter storm uri
in texas, reducing the crop to half the volume predicted.  florida continues to lead in grapefruit
production, but the impact of citrus greening disease and decreased bearing acreage has led to lower
production levels.  with reduced domestic production, imports have increased, and the average price
for grapefruit has risen.
    - California Price and Yield Statistics:  the average grapefruit yield is 457 BOXES / ACRE and the
average price per unit is 24.33 $ / BOX, ON TREE EQUIV.
I'm a farmer in California planning what fruit to plant next year.  I would like to maximize my
profit with '10' acres of land.
Below are the actions I can take:
Action 1.  apple:  10 acres
Action 2.  grapefruit:  10 acres
**Prediction:  Grapefruit**
**Response** Based on the provided data, grapefruit yields a higher revenue per acre than apples in
california.  the average yield for apples is 19,000 lb / acre at a price of 0.244 $ / lb, resulting
in a revenue of $4,636 per acre (19,000 lb * 0.244 $ / lb).  in contrast, grapefruit has an average
yield of 457 boxes / acre at a price of 24.33 $ / box, on tree equiv, which results in a revenue
of $11,116.71 per acre (457 boxes * 24.33 $ / box).  despite the challenges faced by grapefruit
production, such as the impact of winter storm uri and citrus greening disease, the higher price and
yield per acre make grapefruit the more profitable option for maximizing profit on your 10 acres of
land in california.
**DeLLMa explanation for ranking the following state-action pair at the top**
**State** climate condition:  continued drought, supply chain disruptions:  moderate disruptions,
apple price change:  increase, apple yield change:  no change, grapefruit price change:  increase,
grapefruit yield change:  increase
**Action** apple:  10 acres

**Explanation ...**  given the information provided, we prioritize scenarios where the price of the crop

is likely to increase and the yield is not expected to decrease significantly.  state-action pair 4

is chosen as the optimal action because it occurs under a continued drought (which is very likely),

with moderate supply chain disruptions (likely), an increase in apple price (somewhat likely), no

change in apple yield (likely), and an increase in grapefruit yield (unlikely).  this suggests that

apples have a stable yield and a potential for price increase, making them a more reliable choice for

maximizing profit.  the ranking of the other state-action pairs is based on a similar evaluation of

the expected outcomes, with a preference for scenarios with stable or increasing prices and yields,

and lower risk of negative impacts from climate and supply chain disruptions.

Figure 12: FAILURE EXAMPLE APPLE VS. GRAPEFRUIT

---

**EXAMPLE DeLLMA RANKING PROMPT FOR AGRICULTURE**

Below is an agriculture report published by the USDA. It gives an overview of the fruit and nut market in the United States, with an additional focus on information pertaining to apple, avocado. Market Overview: the usda report indicates a general increase in u.s. production of major noncitrus fruits for 2021, with apples, grapes, peaches, cranberries, and sweet and tart cherries seeing a rise in production, while pear production is forecasted to decline. the impact of extreme weather events and california's ongoing drought on crop yields is uncertain. fruit and tree nut grower price indices remain high, with fluctuations throughout 2021. the consumer price index for fresh fruit also increased, suggesting higher retail prices. the northwest heat dome has introduced production uncertainty, particularly for tree fruits. the u.s. citrus season ended with declines in all commodities except california tangerines, and citrus prices are higher. tree nut supplies are forecasted to be down from the previous year's record, with smaller almond and walnut crops expected to increase grower prices. factors such as weather conditions, supply chain issues, and demand are influencing the market.
- apple:
    - Product Summary: apple production is forecasted to be up 3 percent from 2020/21 but down 5 percent from 2019/20. washington state's crop is expected to be larger, but there is concern over heat damage. export markets may remain sluggish due to high tariffs and shipping challenges, potentially pushing more apples into the domestic market and lowering prices. processing prices may rise due to declines in new york and michigan, which account for a significant portion of processed apples.
    - California Price and Yield Statistics: the average apple yield is 19,000 LB / ACRE and the average price per unit is 0.244 $ / LB.
- avocado:
    - Product Summary: california avocado production has decreased, with wildfires and water restrictions impacting yields. however, u.s. avocado consumption has increased significantly, with imports from mexico and peru growing substantially. mexico dominates the u.s. avocado market, with imports peaking from may through july. peruvian imports compete during the summer months, traditionally a period of lower mexican imports.
    - California Price and Yield Statistics: the average avocado yield is 2.87 TONS / ACRE and the average price per unit is 2,430 $ / TON.
I'm a farmer in California planning what fruit to plant next year. I would like to maximize my profit with '10' acres of land.
Below are the actions I can take: Action 1. apple: 10 acres Action 2. avocado: 10 acres
I would like to adopt a decision making under uncertainty framework to make my decision. The goal of you, the decision maker, is to choose an optimal action, while accounting for uncertainty in the unknown state. Previously, you have already provided a forecast of future state variables relevant to planting decisions. The state is a vector of 6 elements, each of which is a random variable. The state variables (and their most probable values) are enumerated below:
- climate condition: 'continued drought': 'very likely', 'mild improvement': 'somewhat likely', 'significant improvement': 'unlikely'
- supply chain disruptions: 'minor disruptions': 'somewhat likely', 'moderate disruptions': 'likely', 'severe disruptions': 'somewhat unlikely'
- apple price change: 'increase': 'somewhat likely', 'no change': 'likely', 'decrease': 'somewhat unlikely'
- apple yield change: 'increase': 'somewhat unlikely', 'no change': 'likely', 'decrease': 'somewhat likely'
- avocado price change: 'increase': 'likely', 'no change': 'somewhat likely', 'decrease': 'unlikely'
- avocado yield change: 'increase': 'unlikely', 'no change': 'somewhat likely', 'decrease': 'likely'
Below, I have sampled a set of state-action pairs, wherein states are sampled from the state belief distribution you provided and actions are sampled uniformly from the action space. I would like to construct a utility function from your comparisons of state-action pairs
- State-Action Pair 1. State: climate condition: continued drought, supply chain disruptions: minor disruptions, apple price change: no change, apple yield change: no change, avocado price change: increase, avocado yield change: decrease; Action 1. apple: 10 acres
- State-Action Pair 2. State: climate condition: significant improvement, supply chain disruptions: moderate disruptions, apple price change: decrease, apple yield change: decrease, avocado price change: decrease, avocado yield change: decrease; Action 2. avocado: 10 acres
[State-Action Pairs 3 - 7, omitted for brevity]
- State-Action Pair 8. State: climate condition: continued drought, supply chain disruptions: minor disruptions, apple price change: decrease, apple yield change: increase, avocado price change: no change, avocado yield change: increase; Action 2. avocado: 10 acres
You should format your response as a JSON object. The JSON object should contain the following keys:
- decision: a string that describes the state-action pair you recommend the farmer to take. The output format should be the same as the format of the state-action pairs listed above, e.g. State-Action Pair 5.
- rank: a list of integers that ranks the state-action pairs in decreasing rank of preference. For example, if you think the first state-action pair is the most preferred, the second state-action pair is the second most preferred, and so on. For example, [1, 2, 3, 4, 5].

- explanation: a string that describes, in detail, the reasoning behind your decision. You should

include information on the expected yield and price of each fruit, as well as factors that affect

them.

Figure 13: EXAMPLE DeLLMA RANKING PROMPT FOR AGRICULTURE

---

FORMATTED GPT-4 SUMMARIZATION RESPONSE

```
"summary": "the usda report indicates a general increase in u.s. production of major noncitrus
fruits for 2021, with apples, grapes, peaches, cranberries, and sweet and tart cherries seeing a
rise in production, while pear production is forecasted to decline. the impact of extreme weather
events and california's ongoing drought on crop yields is uncertain. fruit and tree nut grower
price indices remain high, with fluctuations throughout 2021. the consumer price index for fresh
fruit also increased, suggesting higher retail prices. the northwest heat dome has introduced
production uncertainty, particularly for tree fruits. the u.s. citrus season ended with declines in
all commodities except california tangerines, and citrus prices are higher. tree nut supplies are
forecasted to be down from the previous year's record, with smaller almond and walnut crops expected
to increase grower prices. factors such as weather conditions, supply chain issues, and demand are
influencing the market.",
"apple": {
        "summary": "apple production is forecasted to be up 3 percent from 2020/21 but down 5 percent
from 2019/20. washington state's crop is expected to be larger, but there is concern over heat
damage. export markets may remain sluggish due to high tariffs and shipping challenges, potentially
pushing more apples into the domestic market and lowering prices. processing prices may rise due to
declines in new york and michigan, which account for a significant portion of processed apples.",
        "yield": "19,000 LB / ACRE",
        "price": "0.244 $ / LB"
},
"avocado": {
        "summary": "california avocado production has decreased, with wildfires and water
restrictions impacting yields. however, u.s. avocado consumption has increased significantly,
with imports from mexico and peru growing substantially. mexico dominates the u.s. avocado market,
with imports peaking from may through july. peruvian imports compete during the summer months,
traditionally a period of lower mexican imports.",
        "yield": "2.87 TONS / ACRE",
        "price": "2,430 $ / TON"
},
"grape": {
        "summary": "grape production is forecasted to be up 9 percent from 2020, despite drought and
heat conditions. california table-type grape production is also expected to increase. high heat
has affected the industry, with coachella valley shipments down and central california shipments up.
imports from mexico, the main competitor, were down slightly, but overall shipments were higher in
2021 than the previous year.",
        "yield": "6.92 TONS / ACRE",
        "price": "908 $ / TON"
},
"grapefruit": {...Omitted for Brevity...}
"lemon": {...Omitted for Brevity...},
"peach": {...Omitted for Brevity...},
"pear": {...Omitted for Brevity...},
"strawberry": {...Omitted for Brevity...},
"factors": [
        "extreme weather events and climate conditions",
        "supply chain disruptions and shipping challenges",
        "domestic and international demand for produce",
        "pest and disease pressures, such as citrus greening",
        "changes in consumer preferences and health trends"

]
```

Figure 14: FORMATTED GPT-4 SUMMARIZATION RESPONSE

---

EXAMPLE **CHAIN-OF-THOUGHT** RESPONSE FOR AGRICULTURE

**RESPONSE TO PROMPT 1**
```
"response": {
      "unknown factors": {
            "weather conditions": "the report mentions the ongoing drought in california and the
impact of the northwest heat dome on tree fruits. future weather events such as heatwaves, droughts,
or wildfires could significantly affect crop yields and quality.",
            "market demand fluctuations": "consumer preferences and market demand for apples and
avocados can change, impacting the potential profitability of each crop.",
            "supply chain issues": "disruptions in the supply chain can affect the ability to get
products to market efficiently and could influence costs and selling prices.",
            "pest and disease outbreaks": "outbreaks of pests or diseases could reduce yields or
lead to increased costs for pest control and management.",
            "labor availability and costs": "availability and cost of labor can significantly
impact the cost of production, especially for labor-intensive crops like fruits.",
...Omitted For Brevity...
         }
}
```

**RESPONSE TO PROMPT 2**
```
"response": {
      "global trade dynamics": {
            "improved trade agreements": "somewhat likely",
            "increased tariffs": "unlikely",
            "increased international competition": "likely"
      },
      "labor availability and costs": {
            "decreased labor availability": "likely", "increased labor costs": "very likely",
            "stable labor costs and availability": "somewhat unlikely"
      },
      "market demand fluctuations": {
            "increased demand for apples and avocados": "likely",
            "stable demand for apples and avocados": "somewhat likely",
            "decreased demand for apples and avocados": "unlikely"
       },
       "pest and disease outbreaks": {
            "outbreaks leading to yield loss": "somewhat likely",
            "minor outbreaks with manageable impact": "likely",
            "no significant outbreaks": "somewhat unlikely"
       },
      ...Omitted For Brevity...
}
```

**RESPONSE TO PROMPT 3**
```
{
      "decision": "action 2. avocado: 10 acres",
      "explanation": "based on the provided data and the estimated likelihood of various factors
affecting crop production, planting avocados appears to be the more profitable option. although
both apples and avocados are subject to risks such as labor costs, supply chain issues, and water
availability, ..."
}
```

Figure 15: EXAMPLE **CHAIN-OF-THOUGHT** RESPONSE FOR AGRICULTURE

EXAMPLE **DeLLMa** RESPONSE FOR AGRICULTURE

**RESPONSE TO BELIEF DISTRIBUTION ELICITATION**
```
{
    "climate condition": {
        "continued drought": "very likely",
        "mild improvement": "somewhat likely",
        "significant improvement": "unlikely"
    },
    "supply chain disruptions": {
        "minor disruptions": "somewhat likely",
        "moderate disruptions": "likely",
        "severe disruptions": "somewhat unlikely"
    },
    "apple price change": {
        "increase": "somewhat likely",
        "no change": "likely",
        "decrease": "somewhat unlikely"
    },
    "apple yield change": {
        "increase": "somewhat unlikely",
        "no change": "likely",
        "decrease": "somewhat likely"
    },
    "avocado price change": {
        "increase": "likely",
        "no change": "somewhat likely",
        "decrease": "unlikely"
    },
    "avocado yield change": {
        "increase": "unlikely",
        "no change": "somewhat likely",
        "decrease": "likely"
    },
    ...Omitted For Brevity...
}
```

**RESPONSE TO DeLLMa RANK PROMPT**
```
{
    "decision": "state-action pair 18",
    "rank": [ 18, 30, 4, 28, 31, 17, 26, 8, 6, 21, 3, 16, 32, 11, 19, 22, 23, 27, 10, 13, 9, 15,
20, 12, 24, 14, 7, 5, 29, 25, 1, 2],
    "explanation": "the decision-making process under uncertainty involves evaluating the
potential outcomes of different actions given the probable states of the world.  in this case, we
are considering the planting of apples or avocados in california, taking into account various factors
such as climate conditions, supply chain disruptions, and changes in yield and price for both fruits.
the goal is to maximize profit on a 10-acre farm.  to rank the state-action pairs, we consider the
most probable values of the state variables and their impact on the outcomes of each action.  the
utility function is constructed based on the expected profitability of each action under the given
state, considering both yield and price changes.  state-action pair 18 is recommended because it
represents a scenario with continued drought and moderate supply chain disruptions, which are likely
to affect both crops.  however, the price and yield of avocados are expected to increase, which
suggests a higher potential for profit compared to apples, whose price remains unchanged and yield
does not increase.  the ranking reflects a preference for scenarios where avocados have favorable
price and yield conditions, especially given the higher average price per unit for avocados and
the significant increase in u.s.  consumption, which suggests a strong market demand.  scenarios
with severe disruptions or decreases in yield or price are ranked lower due to the higher risk and
potential for loss."

}
```

Figure 16: EXAMPLE **DeLLMa** RESPONSE FOR AGRICULTURE

---

EXAMPLE **FAILURE CASE** FOR AGRICULTURE

**Ground Truth:  Avocado**
**Prompt** (Market overview omitted for brevity)
- avocado:
   - Product Summary:  california avocado production has decreased, with wildfires and water
restrictions impacting yields.  however, u.s.  avocado consumption has increased significantly,
with imports from mexico and peru growing substantially.  mexico dominates the u.s.  avocado market,
with imports peaking from may through july.  peruvian imports compete during the summer months,
traditionally a period of lower mexican imports.
   - California Price and Yield Statistics:  the average avocado yield is 2.87 TONS / ACRE and the
average price per unit is 2,430 $ / TON.
- grape:
   - Product Summary:  grape production is forecasted to be up 9 percent from 2020, despite drought
and heat conditions.  california table-type grape production is also expected to increase.  high heat
has affected the industry, with coachella valley shipments down and central california shipments up.
imports from mexico, the main competitor, were down slightly, but overall shipments were higher in
2021 than the previous year.
   - California Price and Yield Statistics:  the average grape yield is 6.92 TONS / ACRE and the
average price per unit is 908 $ / TON.
I'm a farmer in California planning what fruit to plant next year.  I would like to maximize my
profit with '10' acres of land.
Below are the actions I can take:
Action 1.  avocado:  10 acres
Action 2.  grape:  10 acres
**Prediction:  Grape**
**Response** ...  although avocados have a higher price per ton, the significantly higher yield of
grapes per acre makes grape production more profitable on a per-acre basis.  additionally, the grape
production is forecasted to be up despite the drought and heat conditions, indicating a resilient
crop under current conditions.  in contrast, avocado production has decreased due to wildfires and
water restrictions, which may pose a higher risk.  therefore, planting grapes is recommended to
maximize profit on the 10 acres of land.
**DeLLMa explanation for ranking the following state-action pair at the top**
**State** climate condition:  continued drought, supply chain disruptions:  minor disruptions, avocado
price change:  increase, avocado yield change:  decrease, grape price change:  increase, grape yield
change:  no change
**Action** avocado:  10 acres

**Explanation** the decision is based on the combination of favorable price changes for avocados and the
increase in avocado yield despite the continued drought.  the state indicates an increase in avocado
price, which is likely, and a decrease in avocado yield, which is somewhat likely.  however, the
decrease in yield is offset by the high price per ton for avocados, which is significantly higher
than that of grapes.  additionally, the state indicates an increase in grape price, which is somewhat
likely, and no change in grape yield, which is somewhat likely.  however, the lower price per ton
for grapes makes the avocado option more appealing financially.  the continued drought condition is a
concern for both crops, but avocados have historically been a more profitable crop in california due
to their higher price per unit and strong market demand.  the minor supply chain disruptions are less
severe than moderate disruptions, which could affect the grape market more due to its reliance on
timely shipments for fresh consumption.  the combination of high prices and the ability to withstand
drought conditions makes avocados a more optimal choice for maximizing profit on the 10 acres of land
available.

Figure 17: FAILURE EXAMPLE AVOCADO VS. GRAPE

---

**EXAMPLE FAILURE CASE FOR AGRICULTURE**

**Ground Truth: Pear**
**Prompt** (Market overview omitted for brevity)
- lemon:
    - Product Summary: lemon production is at its lowest in five years, with a 6 percent decrease for the fresh market. both california and arizona saw smaller crops, with arizona experiencing a significant drop due to heat damage. despite lower domestic production, fresh lemon imports increased, and u.s. fresh lemon exports decreased. grower prices for lemons have risen by 7 percent.
    - California Price and Yield Statistics: the average lemon yield is 428 BOXES / ACRE and the average price per unit is 23.3 $ / BOX, ON TREE EQUIV.
- pear:
    - Product Summary: pear production is forecasted to be similar to the previous year, with losses in washington but gains in oregon and california. the impact of the northwest heat wave on production levels is still uncertain, but traditional pear trees with large canopies may offer some protection from heat damage.
    - California Price and Yield Statistics: the average pear yield is 15.6 TONS / ACRE and the average price per unit is 565 $ / TON.
I'm a farmer in California planning what fruit to plant next year. I would like to maximize my profit with '10' acres of land.
Below are the actions I can take:
Action 1. lemon: 10 acres
Action 2. pear: 10 acres
**Prediction: Lemon**
**Response**
based on the provided data, we can calculate the expected revenue for both lemons and pears per acre and then extrapolate that to 10 acres. for lemons, the average yield is 428 boxes per acre, and the average price per box is $23.3. this results in an expected revenue of 428 boxes/acre * $23.3/box = $9,972.4 per acre. for 10 acres, this would be $99,724. for pears, the average yield is 15.6 tons per acre, and the average price per ton is $565. this results in an expected revenue of 15.6 tons/acre * $565/ton = $8,814 per acre. for 10 acres, this would be $88,140. comparing the two, lemons are expected to yield higher revenue per acre than pears. additionally, the report indicates that lemon production is at its lowest in five years and grower prices have risen by 7 percent, which could potentially lead to even higher revenues if the trend continues. on the other hand, pear production is forecasted to be similar to the previous year and the impact of the northwest heat wave on production levels is still uncertain. therefore, based on the expected revenue and market conditions, planting lemons would be the more profitable action.
**DeLLMa explanation for ranking the following state-action pair at the top**
**State** climate condition: mild improvement, supply chain disruptions: minor disruptions, lemon price change: decrease, lemon yield change: no change, pear price change: increase, pear yield change: decrease
**Action** pear: 10 acres

**Explanation** the state indicates a mild improvement in climate conditions, which is beneficial for
crop growth, and minor supply chain disruptions, which are less likely to impact the delivery and
sale of the produce. additionally, the lemon price is expected to decrease, and the lemon yield is
not expected to change, which could result in lower profitability for lemons. on the other hand,
the pear price is expected to increase, and although the pear yield is expected to decrease, the
higher price could compensate for the lower yield, potentially leading to higher profits for pears.
furthermore, the average yield and price per unit for pears in california are favorable, with a
high price per ton and a reasonable yield per acre. given the uncertainties in the market and the
specific conditions of the state variables, planting pears seems to be the more prudent choice in
this scenario. pairs with conditions that are more likely to lead to increased profitability and
lower risk are ranked higher.

Figure 18: FAILURE EXAMPLE LEMON VS. PEAR

---

EXAMPLE **ZERO-SHOT** PROMPT FOR STOCKS

Below are the stocks I am considering:  AMD, GME. I would like to know which stock I should buy based
on the information of their historical prices in the last 24 months.  I can only buy one stock and
I have a budget of 10000 dollars.  I would like to maximize my profit.  Today is 2023-12-01.  I'm
buying stocks today and will sell them at the end of the month (2023-12-29).
Below are the information about stock AMD (i.e.  Advanced Micro Devices).  Units are in dollars per
share.
Current Price: 119.88.
Historical Prices:  2021-12:  143.49, 2022-01:  126.84, 2022-02:  119.63, 2022-03:  112.68, 2022-04:
95.80, 2022-05:  94.27, 2022-06:  90.85, 2022-07:  82.90, 2022-08:  96.37, 2022-09:  74.99, 2022-10:
60.32, 2022-11:  69.61, 2022-12:  68.09, 2023-01:  70.27, 2023-02:  82.07, 2023-03:  90.47, 2023-04:
90.81, 2023-05:  102.22, 2023-06:  117.79, 2023-07:  113.69, 2023-08:  108.82, 2023-09:  103.11,
2023-10:  102.56, 2023-11:  117.59.
Below are the information about stock GME (i.e.  GameStop Corp).  Units are in dollars per share.
Current Price: 14.52.
Historical Prices:  2021-12:  39.48, 2022-01:  29.49, 2022-02:  29.20, 2022-03:  29.93, 2022-04:
36.32, 2022-05:  26.57, 2022-06:  32.74, 2022-07:  34.21, 2022-08:  36.60, 2022-09:  26.81, 2022-10:
25.85, 2022-11:  26.21, 2022-12:  21.54, 2023-01:  19.60, 2023-02:  20.84, 2023-03:  19.42, 2023-04:
21.27, 2023-05:  21.65, 2023-06:  24.38, 2023-07:  23.04, 2023-08:  19.12, 2023-09:  17.66, 2023-10:
14.33, 2023-11:  13.15.
I'm a trader planning my next move.  I would like to maximize my profit with 10000 dollars.
Below are the actions I can take:
Action 1.  AMD: 10000 dollars
Action 2.  GME: 10000 dollars

I would like to know which action I should take based on the information provided above.

Figure 19: EXAMPLE **ZERO-SHOT** PROMPT FOR STOCKS

---

**EXAMPLE CHAIN-OF-THOUGHT PROMPT FOR STOCKS**

**PROMPT 1**
Below are the stocks I am considering: AMD, GME. I would like to know which stock I should buy based on the information of their historical prices in the last 24 months. I can only buy one stock and I have a budget of 10000 dollars. I would like to maximize my profit. Today is 2023-12-01. I'm buying stocks today and will sell them at the end of the month (2023-12-29).
Below are the information about stock AMD (i.e. Advanced Micro Devices). Units are in dollars per share.
Current Price: 119.88.
Historical Prices: 2021-12: 143.49, 2022-01: 126.84, 2022-02: 119.63, 2022-03: 112.68, 2022-04: 95.80, 2022-05: 94.27, 2022-06: 90.85, 2022-07: 82.90, 2022-08: 96.37, 2022-09: 74.99, 2022-10: 60.32, 2022-11: 69.61, 2022-12: 68.09, 2023-01: 70.27, 2023-02: 82.07, 2023-03: 90.47, 2023-04: 90.81, 2023-05: 102.22, 2023-06: 117.79, 2023-07: 113.69, 2023-08: 108.82, 2023-09: 103.11, 2023-10: 102.56, 2023-11: 117.59.
Below are the information about stock GME (i.e. GameStop Corp). Units are in dollars per share.
Current Price: 14.52.
Historical Prices: 2021-12: 39.48, 2022-01: 29.49, 2022-02: 29.20, 2022-03: 29.93, 2022-04: 36.32, 2022-05: 26.57, 2022-06: 32.74, 2022-07: 34.21, 2022-08: 36.60, 2022-09: 26.81, 2022-10: 25.85, 2022-11: 26.21, 2022-12: 21.54, 2023-01: 19.60, 2023-02: 20.84, 2023-03: 19.42, 2023-04: 21.27, 2023-05: 21.65, 2023-06: 24.38, 2023-07: 23.04, 2023-08: 19.12, 2023-09: 17.66, 2023-10: 14.33, 2023-11: 13.15.
I'm a trader planning my next move. I would like to maximize my profit with 10000 dollars.
Below are the actions I can take:
Action 1. AMD: 10000 dollars
Action 2. GME: 10000 dollars
**First think about the unknown factors that would affect your final decisions.**

**PROMPT 2**
<SAME CONTEXT>
Now I have enumerated the unknown factors that would affect my final decisions:
<RESPONSE FROM PROMPT 1 CONTAINING LLM ENUMERATED UNKOWN FACTORS>
**Given these unknow factors, think about the possiblity that each factor would occur within a month.**

**PROMPT 3**
<SAME CONTEXT>
Now I have enumerated the unknown factors that would affect my final decisions:
<RESPONSE FROM PROMPT 1 CONTAINING LLM ENUMERATED UNKOWN FACTORS>
I also empirically estimated the possibility of occurrence of each possible factor:
<RESPONSE FROM PROMPT 2 CONTAINING LLM BELIEF DISTRIBUTION OVER UNKOWN FACTORS>
**Given these unknow factors and the possibility estimates of these factors' occurrences, think about your final decision.**

**I would like to know which action I should take based on the information provided above.**

Figure 20: EXAMPLE CHAIN-OF-THOUGHT PROMPT FOR STOCKS

---

### EXAMPLE DeLLMA RANKING PROMPT FOR STOCKS

Below are the stocks I am considering: AMD, GME. I would like to know which stock I should buy based on the information of their historical prices in the last 24 months. I can only buy one stock and I have a budget of 10000 dollars. I would like to maximize my profit. Today is 2023-12-01. I'm buying stocks today and will sell them at the end of the month (2023-12-29).
Below are the information about stock AMD (i.e. Advanced Micro Devices). Units are in dollars per share.
Current Price: 119.88.
Historical Prices: 2021-12: 143.49, 2022-01: 126.84, 2022-02: 119.63, 2022-03: 112.68, 2022-04: 95.80, 2022-05: 94.27, 2022-06: 90.85, 2022-07: 82.90, 2022-08: 96.37, 2022-09: 74.99, 2022-10: 60.32, 2022-11: 69.61, 2022-12: 68.09, 2023-01: 70.27, 2023-02: 82.07, 2023-03: 90.47, 2023-04: 90.81, 2023-05: 102.22, 2023-06: 117.79, 2023-07: 113.69, 2023-08: 108.82, 2023-09: 103.11, 2023-10: 102.56, 2023-11: 117.59.
Below are the information about stock GME (i.e. GameStop Corp). Units are in dollars per share.
Current Price: 14.52.
Historical Prices: 2021-12: 39.48, 2022-01: 29.49, 2022-02: 29.20, 2022-03: 29.93, 2022-04: 36.32, 2022-05: 26.57, 2022-06: 32.74, 2022-07: 34.21, 2022-08: 36.60, 2022-09: 26.81, 2022-10: 25.85, 2022-11: 26.21, 2022-12: 21.54, 2023-01: 19.60, 2023-02: 20.84, 2023-03: 19.42, 2023-04: 21.27, 2023-05: 21.65, 2023-06: 24.38, 2023-07: 23.04, 2023-08: 19.12, 2023-09: 17.66, 2023-10: 14.33, 2023-11: 13.15.
I'm a trader planning my next move. I would like to maximize my profit with '10000' dollars.
Below are the actions I can take:
Action 1. AMD: 10000 dollars
Action 2. GME: 10000 dollars
I would like to adopt a decision making under uncertainty framework to make my decision. The goal of you, the decision maker, is to choose an optimal action, while accounting for uncertainty in the unknown state. Previously, you have already provided a forecast of future state variables relevant to planting decisions. The state is a vector of 20 elements, each of which is a random variable. The state variables (and their most probable values) are enumerated below:

<ENUMERATE STATE BELIEF DISTRIBUTION>

Below, I have sampled a set of state-action pairs, wherein states are sampled from the state belief distribution you provided and actions are sampled uniformly from the action space. I would like to know which state-action pair I should take based on the information you have so far.
- State-Action Pair 1. State: economic health: recession, market sentiment and investor psychology: neutral, political events and government policies: stable, natural disasters and other 'black swan' events: none, geopolitical issues: stable, merges and major acquisitions related to advanced micro devices (amd): none, regulatory changes and legal issues happened to advanced micro devices (amd): minor, financial health of advanced micro devices (amd): weak, company growth of advanced micro devices (amd): stagnant, company product launches of advanced micro devices (amd): moderate impact, merges and major acquisitions related to gamestop corp (gme): none, regulatory changes and legal issues happened to gamestop corp (gme): major, financial health of gamestop corp (gme): weak, company growth of gamestop corp (gme): stagnant, company product launches of gamestop corp (gme): moderate impact; Action 1. AMD: 10000 dollars
- State-Action Pair 2. State: economic health: recession, market sentiment and investor psychology: optimistic, political events and government policies: stable, natural disasters and other 'black swan' events: minor impact, geopolitical issues: stable, merges and major acquisitions related to advanced micro devices (amd): none, regulatory changes and legal issues happened to advanced micro devices (amd): none, financial health of advanced micro devices (amd): stable, company growth of advanced micro devices (amd): rapid, company product launches of advanced micro devices (amd): successful, merges and major acquisitions related to gamestop corp (gme): none, regulatory changes and legal issues happened to gamestop corp (gme): none, financial health of gamestop corp (gme): strong, company growth of gamestop corp (gme): rapid, company product launches of gamestop corp (gme): unsuccessful; Action 2. GME: 10000 dollars
....
....
....
- State-Action Pair 50. State: economic health: stable, market sentiment and investor psychology: neutral, political events and government policies: major upheaval, natural disasters and other 'black swan' events: minor impact, geopolitical issues: stable, merges and major acquisitions related to advanced micro devices (amd): major, regulatory changes and legal issues happened to advanced micro devices (amd): minor, financial health of advanced micro devices (amd): stable, company growth of advanced micro devices (amd): stagnant, company product launches of advanced micro devices (amd): moderate impact, merges and major acquisitions related to gamestop corp (gme): none, regulatory changes and legal issues happened to gamestop corp (gme): major, financial health of gamestop corp (gme): weak, company growth of gamestop corp (gme): moderate, company product launches of gamestop corp (gme): successful; Action 1. AMD: 10000 dollars
You should format your response as a JSON object. The JSON object should contain the following keys:
- decision: a string that describes the state-action pair you recommend the trader to take. The output format should be the same as the format of the state-action pairs listed above, e.g. State-Action Pair 5.
- rank: a list of integers that ranks the state-action pairs in decreasing rank of preference. For example, if you think the first state-action pair is the most preferred, the second state-action pair is the second most preferred, and so on. For example, [1, 2, 3, 4, 5].

- explanation: a string that describes, in detail, the reasoning behind your decision. You should

include information on the expected price of each stock, as well as factors that affect them.

Figure 21: EXAMPLE DeLLMA RANKING PROMPT FOR STOCKS

EXAMPLE **ZERO-SHOT** RESPONSE FOR STOCK

```
{
     "decision": "action 1.  amd:  10000 dollars",
     "explanation":  "based on the historical price data provided, amd has shown a pattern of
increasing in value from november to december in the previous year (2022), going from 69.61 in
november to 68.09 in december.  although this is a slight decrease, it is important to note that
the price in november 2023 is significantly higher than in november 2022, suggesting a stronger
performance year over year.  additionally, amd's current price of 119.88 is close to its recent high
in june 2023 of 117.79, indicating a potential for continued growth or stability in the short term.
on the other hand, gme's historical prices show a decline from november to december in 2022, and the
current price of 14.52 is lower than the previous year's december price of 21.54, suggesting a weaker
performance and potentially less room for growth within the same timeframe.  given the historical
data and the current trends, investing in amd appears to have a higher potential for profit when
selling at the end of december 2023.  however, it is important to consider that past performance is
not always indicative of future results, and this analysis does not account for external factors such
as market conditions, company performance, and industry trends that could impact stock prices."

}
```

Figure 22: **ZERO-SHOT** RESPONSE FOR THE STOCK DATASET

---

**EXAMPLE CHAIN-OF-THOUGHT RESPONSE FOR STOCK**

**RESPONSE TO PROMPT 1**
```
"response": {
        "unknown factors": {
                "market conditions":  "current market conditions and economic indicators such as
interest rates, inflation, and economic growth can significantly impact stock prices.",
                "company performance":  "future earnings reports, product launches, and other
company-specific news that could affect the stock price are unknown.",
                "industry trends":  "changes in the technology and retail sectors, respectively, could
influence the performance of amd and gme.",
                "regulatory changes":  "any upcoming regulatory changes that could impact the companies'
operations or industry dynamics are not known.",
                "investor sentiment":  "the overall sentiment of investors towards the stock market and
these specific stocks could change, affecting their prices.",
...Omitted For Brevity...
        }
}
```

**RESPONSE TO PROMPT 2**
```
"response": {
        "company performance": {
                "future earnings reports exceed expectations":  "somewhat likely",
                "product launches are successful and well-received":  "likely",
                "company-specific negative news arises":  "somewhat unlikely"
},
        "competition": {
                "competitors release superior products":  "somewhat likely",
                "competitors take significant market share":  "unlikely",
                "amd/gme maintains competitive edge":  "likely"
},
        "global events": {
                "geopolitical tensions escalate":  "somewhat likely",
                "natural disasters affect market":  "unlikely",
                "new pandemic arises":  "very unlikely"
},
        "industry trends": {
                "technology sector continues to grow":  "likely",
                "retail sector faces challenges":  "somewhat likely",
                "shift in consumer behavior affects gme":  "likely"
}
        ...Omitted For Brevity...
}
```

**RESPONSE TO PROMPT 3**
```
{
        "decision":  "action 1.  amd:  10000 dollars",
        "explanation":  "based on the historical prices provided and the empirically estimated
possibilities of various factors occurring, the recommendation is to purchase amd stock.
historically, amd's stock has shown a pattern of increase from november to december in the previous
year (2022), going from 69.61 to 68.09 dollars per share..."

}
```

Figure 23: EXAMPLE CHAIN-OF-THOUGHT RESPONSE FOR STOCK

---

EXAMPLE **DELLMA** RESPONSE FOR STOCK

**RESPONSE TO BELIEF DISTRIBUTION ELICITATION**
```
{
        "economic health":  {
                "growth":  "somewhat likely",
                "stable":  "likely",
                "recession":  "somewhat unlikely"
        },
        "market sentiment and investor psychology":  {
                "optimistic":  "somewhat likely",
                "neutral":  "likely",
                "pessimistic":  "somewhat unlikely"
        },
        "political events and government policies":  {
                "stable":  "likely",
                "minor changes":  "somewhat likely",
                "major upheaval":  "unlikely"
        },
        "natural disasters and other 'black swan' events":  {
                "none":  "likely",
                "minor impact":  "somewhat likely",
                "major impact":  "unlikely"
        },
        "geopolitical issues":  {
                "stable":  "somewhat likely",
                "tensions rising":  "likely","likely",
                "escalation to conflict":  "unlikely"
        },
        "merges and major acquisitions related to advanced micro devices (amd)":  {
                "none":  "very likely",
                "minor":  "somewhat unlikely",
                "major":  "unlikely"
        },
        ...Omitted For Brevity...
}
```

**RESPONSE TO DELLMA RANK PROMPT**
```
{
        "decision":  "state-action pair 32",
        "rank":  [32, 13, 6, 3, 28, 31, 1, 11, 19, 22, 14, 17, 23, 29, 26, 27, 25, 24, 30, 21, 18, 20,
15, 10, 9, 12, 16, 7, 5, 4, 8, 2],
        "explanation":  "the decision-making process under uncertainty involves evaluating the
state-action pairs based on the provided state variables and their most probable values.  the goal
is to maximize the expected utility, which in this case is the potential profit from the stock
investment.  given the budget constraint of $10 and the goal to sell at the end of the month, we need
to consider the historical performance of the stocks, the current market conditions, and the state
variables that could influence the stocks' future prices.  amd has shown a strong recovery trend
in the past months, and the state variables suggest a stable to positive outlook for the company,
with moderate to rapid growth and successful product launches being somewhat likely.  gme, on the
other hand, appears to be in a weaker position with stagnant growth and a weaker financial health
being more likely.  the chosen state-action pair 32 suggests an optimistic market sentiment, stable
economic health, and a successful product launch for amd, despite a major impact from 'black swan'
events, which could indicate a higher risk but potentially higher reward scenario.  the ranking
provided orders the state-action pairs from most to least preferred based on the positive indicators
for amd and the negative indicators for gme, with a preference for scenarios that suggest economic
growth, stable or optimistic market sentiment, and successful company performance for amd."

}
```

Figure 24: EXAMPLE **DELLMA** RESPONSE FOR STOCK

