# OpenReview forum: "DeLLMa: Decision Making Under Uncertainty with Large Language Models"
_ICLR.cc/2025/Conference — ICLR 2025 Spotlight_

### Official Review · Reviewer_GCWw · 2024-10-29

**Soundness:** 4
**Presentation:** 3
**Contribution:** 3
**Rating:** 8
**Confidence:** 4

**Summary:**

The paper introduces DeLLMa for better decision making by LLMs under uncertainty. It uses a multistep process to achieve this, based on decision theory - state enumeration, forecasting, then utility elicitation and expectation maximization. Then the paper applies this to agro and fin scenarios - showing good improvement over baselines (zero shot, CoT etc). Finally the paper shows that this method generalizes across LLMs and is human auditable.

**Strengths:**

Originality - the paper is quite original, and the proposed pipeline is creative. I have not seen this multistep approach used with LLMs anywhere before.

Quality - I found the method to be quite well motivated and well presented. The paper is reproducible, and the experiments seem cohesive and well-motivated.

Clarity - The paper is well articulated and clearly structured.

Significance - The paper is definitely an important step in the right direction. Uncertainity quantification in ML has been an active area of research and I am curious to see how this is furthered with LLMs. However I feel that the method has limited applicability to real world scenarios.

**Weaknesses:**

1. The context window of LLMs clearly limits the state and action spaces, which makes this method of limited real world applicability.
2. It appears that the method is very sensitive to the first state forecasting step. While the method is robust in general, I would like to understand if this is indeed a big bottleneck.
3. The utility elicitaiton method proposed (pairwise ranking) seems a bit too simple for real world scenarios. I would like to know if other methods were tried for this.

**Questions:**

1. How can DeLLMa be adapted to handle continuous state and action spaces?
2. The state forecasting method assumes independence bw latent factors. How does this impact total performance?
3. Is DeLLMa able to handle cases where the user task cannot be distilled into a hard utility function?

---

> ### Author Response · Authors · 2024-11-22
> **Official Comment by Authors**
>
> Thank you so much for your positive feedback. We are delighted that you found DeLLMa to be original, clearly presented, and achieve competitive performance. We hope to address your questions and concerns below.
>
> ### __Context Window Limits__
>
> We agree that the context window may indeed pose some limits on real world problems. Nonetheless, most leading LLMs are endowed with a context window of 128K tokens, and our most context-intensive instances only require <21K tokens as contexts (shown in Table 5). We believe that this window, together with future advances in long-context modeling, can cover a broad range of decision making problems.
>
> ### __Sensitivity to State Forecasts__
>
> We did observe some sensitivity to the state forecasting step. But as we show in Table 2 of our paper—which shows DeLLMa performance given different state forecasting variations—DeLLMa seems to be quite robust to this step as long as the backbone LLM can perform the utility elicitation step well.
>
>
> ### __Simplicity of Pairwise Ranking__
>
> We have indeed tried a number of variations for utility elicitation! For human pairwise comparisons, we tried variants such as direct rankings and rating scales (from a range of 1 to 5). For utility fitting, we also tried different utility fitting methods provided in the `choix` package [1], such as the iterative variant of the Luce-Spectral Ranking method. We found that the methods described in section 3.3 worked the best for the types of problems we have in the experiments. However, as we expand the problem space in the future, we will be sure to evaluate other methods.
>
> ### __Reviewer Questions__
>
> > How can DeLLMa be adapted to handle continuous state and action spaces?
>
> This is a great question, which we are exploring:
>
> - __[Continuous State Space]__: There are multiple possibilities for this—one direction we are exploring is tool usage (where we use probabilistic tools to yield a continuous state forecast), but we can also directly prompt an LLM to forecast a distribution over the continuous state variables.
>
> - __[Continuous Action Space]__: For a continuous action space, we just need to adapt/update our expected utility maximization step to a continuous space. We can offload this to a suitable continuous optimization algorithm or program. For example, we can perform adaptive gradient steps over the action space when optimizing continuous actions. (c.f.  Section 4.7.2 of [2])
>
> > The state forecasting method assumes independence bw latent factors. How does this impact total performance?
>
> It’s a good point that the independence assumption will affect performance. While we show that this assumption works reasonably in practice in our current implementation (e.g., Table 1), we haven’t yet implemented a more general probabilistic model to assess the improvement—but it’s an important direction which we will do in future work.
>
> > Is DeLLMa able to handle cases where the user task cannot be distilled into a hard utility function?
>
> In our current implementation, we assume that enough information to specify a utility function is given by the user prompt. But in the future we will definitely aim to address cases where the utility is only partially defined (or where there is ambiguity). Some prior works (e.g. [3]) have shown that preference elicitation from limited information is indeed possible.
>
> [1] The `choix` package, https://choix.lum.li/en/latest/api.html#process-pairwise
>
> [2] Decision Making Under Uncertainty, https://web.stanford.edu/group/sisl/public/dmu.pdf
>
> [3] Eliciting Human Preferences with Language Models, https://arxiv.org/abs/2310.11589

---

### Official Review · Reviewer_4x55 · 2024-11-01

**Soundness:** 2
**Presentation:** 3
**Contribution:** 2
**Rating:** 6
**Confidence:** 4

**Summary:**

This paper proposes DeLLMa (decision-making LLM assistant) to maximise the utility in decision problems, i.e., agriculture planning and finance. DeLLMa consists three steps, 1) based on in-context information, infer and forecast related unknown variables, 2) elicit the utility function which aligns with the user's goals, 3) use the elicited utility function to make the best decision which maximise the expected utility.   Authors claim that DeLLMa is motivated from chain of thoughts and tree of thoughts, but is faster at the inference time especially in the scalability.

**Strengths:**

- Clearly explain the main concept of DeLLMa as illustrated in Figure 1.
- Capture LLM in decision making with a triplet P=(G, A, C) and four steps, where G is the user goal inferred from the description, A is a list of actions, and C is the contextual information.
- Include two ranking types: pairwise and top-1.
- Provide clear prompts and code in the appendix.

**Weaknesses:**

- Inferring the user’s goal from the context is done by the LLM, which requires the user to have a clear goal in mind. For example, if the farmer has not decided to plant only one type of fruit, DeLLMa may not know how to handle an implicit multiple-fruit situation.
- The applicable scope of DeLLMa is limited. The experiments include only agriculture and stocks, which may mislead readers into thinking DeLLMa generalises across all decision-making tasks.
- There is little information about human agreement; the paper states that the authors served as human annotators, but details are missing.

**Questions:**

- How will DeLLMa perform on other tasks related to chain-of-thought or tree-of-thought methods discussed in related work?
- If scalability is not a significant issue in certain cases, will DeLLMa outperform other methods?
- In the experiments, one task involves investing on December 1, 2023, and selling on the last trading day of that month (December 29, 2023) to maximize returns. Why were these specific dates chosen? Why not other dates?

---

> ### Author Response · Authors · 2024-11-22
> **Official Comment by Authors**
>
> Thank you for your feedback! We are glad that you found our formalism helpful in understanding our framework, and our results reproducible. We hope to address your questions and concerns below.
>
> ### __Fixed Decision Space__
>
> This is a great point! Indeed, many real-world decisions do not attain a predefined set of actions, and this is an important future direction that we aim to tackle. Nonetheless, there are many important problems that are endowed with a fixed action set. For example, in healthcare, physicians often make decisions from a finite set of treatment options based on a patient’s symptoms and diagnostic result [1]. Similarly, in supply chain management, decision-makers regularly work within a bounded set of distribution strategies and inventory policies, such as selecting from fixed shipping options or standard order quantities [2]. With a focus on this setting, DeLLMa is our first step towards a more comprehensive system for real-life decision making.
>
> ### __Limited Scope of DeLLMa Applications__
>
> Thank you for raising this concern! It is certainly not our intention to phrase DeLLMa as a general solution to all instances of decision making under uncertainty. In fact, in Appendix H, we emphasize that DeLLMa is only a first step towards this direction, and requires additional guardrails and human-AI collaboration before becoming a production-ready system. In this section in our paper, we have added some additional discussion which reads as follows:
> ```
> Furthermore, our current experiments focus on specific domains (agriculture and stocks) as representative, controlled environments for testing decision-making under uncertainty. Future work will involve evaluating DeLLMa across a broader set of domains to better understand its strengths and limitations in diverse, real-world applications.
> ```
>
> ### __Missing Details on Human Annotation Study__
>
> We appreciate your criticism and apologize for any confusion this may have caused! In our global response, we have provided the annotation guidelines and updated annotation results. In short, we have solicited a total of 412 annotations, 200 of which are annotated by 5 external volunteers. From these, we obtained an annotator-LLM agreement rate of 66.75% and an inter-annotator agreement rate of 67.0\% ($\pm$ 6.34\%). All annotation results and evaluation scripts have been updated to our supplementary material (under `DeLLMa-additional-annotation`).
>
> ### __Reviewer Questions__
>
> > How will DeLLMa perform on other tasks related to chain-of-thought or tree-of-thought methods discussed in related work?
>
> Thank you for this question. DeLLMa is most well suited to decision-making under uncertainty tasks, which is a subset of reasoning tasks. While CoT / ToT are more-broadly applicable to a wider range of tasks, we've found that DeLLMa yields higher performance on these decision making tasks.
>
> > If scalability is not a significant issue in certain cases, will DeLLMa outperform other methods?
>
> Yes, we observe in Figure 3 that DeLLMa does improve with scale, and its scaling behavior compares favorably with OpenAI o1 (shown in Table 3), a SotA method for scaling compute at inference-time aimed at reasoning tasks.
>
> > In the experiments, one task involves investing on December 1, 2023, and selling on the last trading day of that month (December 29, 2023) to maximize returns. Why were these specific dates chosen? Why not other dates?
>
> Our initial experiments were conducted on the GPT-4-1106 checkpoint (i.e., a model released in November of 2023), and we use financial market data from December 2023 for evaluation since they are strictly out of the knowledge cutoff of the model.
>
> [1] MIMIC-IV-Ext Clinical Decision Making: A MIMIC-IV Derived Dataset for Evaluation of Large Language Models on the Task of Clinical Decision Making for Abdominal Pathologies, https://physionet.org/content/mimic-iv-ext-cdm/1.1/
>
> [2] Large Language Models for Supply Chain Optimization, https://arxiv.org/abs/2307.03875

---

> > ### Author Response · Authors · 2024-11-25
> > **Thanks for Your Valuable Input**
> >
> > Dear Reviewer 4x55,
> >
> > Thank you once again for your insightful feedback! We hope our response has addressed your concerns satisfactorily, and we would be happy to engage further if you have any additional questions or suggestions.

---

### Official Review · Reviewer_RA2F · 2024-11-05

**Soundness:** 3
**Presentation:** 3
**Contribution:** 3
**Rating:** 8
**Confidence:** 4

**Summary:**

The authors  propose  framework to work with LLMs for decision making under uncertainty; in doing so, they take classical utility/decision theoretic framework as a design guideline: (1) they enumerate first all the states. (2) estimate the probability of those states (3) elicit the utility function from the user, and (4) choose the one that maximises expected utility.   They show their results working with various LLM and in various settings; different preference ranking strategies (pair by pair vs. top1) against benchmarks of zero shot and CoT in two real world  decision problems (one in agriculture the other in stocks investment) and show that they their framework helps with the results significantly. Also considered are ablation studies (on  things such as overlap percentage (of minitibatches in drawing from state-action samples) and sample size, human agreement etc ). Approach also comes with the decision trees naturally, hence making the whole process more explainable.

**Strengths:**

- Exposition and writing  in general is in  good condition. (There are small rooms for improvement in organisation and clarification.)

- The whole framework is written with careful formal details, thanks to taking decision-theoretic frame, which is a great idea to start with.

-  The results on improvements are powerful; also has high impact potential  due to popularity of prompt engineering in LLMs. The idea is novel and justifiable (with some inherent difficulties and pitfalls ).

- The depth of the analysis and various ablation studies is a strength.

- The framework also provides (to some extent) an explainable medium which is a strength.

**Weaknesses:**

- Content-wise: Unfortunately, too many reference to appendix in many crucial points. This is of course due to  tight page limitation, but authors should seriously think about re-arranging the text more complete. For instance explanations for Table 2 , 3 and 4 really not sufficient (page 10 only call their names without telling what they are).  It seems like the paper would serve much better as a journal publication or other venues with more page (e.g., ICML).  Related: Say also that Figure 11  is in appendix actually.  They try to provide too much (for instance human evaluation at best a study on its own, or should be at least extended to present something substantial)

- On the technical side there are strong assumptions and unclarities:  the assumption of independence between k-latent factors is very strong, and  actually hard to maintain,  also due to natural language part of things.  As a result, uncertainty in the estimation step is taken very crisp, it is pretty difficult to ensure whether these scores are reflecting reality. There is a section called: "How good are state forecasts" (which is a good to have section), however I am not sure it reaches its goal:  This is inherently difficult. They show table 1: claiming two metrics they showing "good results" is hard to justify since we don't know reference to what (maybe its my misunderstanding.) Why are these reasonable? Also the same section, manually annotating a set of ground truth values, how to get it, etc, not very clear to me.

- Less problematic is the issue of  the number of alternatives are known (coming from the context or imposed to LLM) is a strong working assumption, and undermines  real world settings applicability.

-Another major issue is the part with Table 4 (human evaluation): this is a nice to have I agree, however, if you go for it then it should  be done in a state that does not weaken the paper. Currently, as explain this is just done by authors, of which we don't know how many, say 4 or 5 on average. Is this result than solid? It is problematic for two reasons: 1)  It should be done not by authors  but by a handful of people (who has nothing to do with paper acceptance) 2) not clear how does it compare to sole human comparison 3) amount of people matters: what if it was a single author paper (I assume it is not due to their expression). What is the standard deviation, significance level  or other statistics on these values? Without all these things , it would be pretty difficult to publish this in even a low-rank psychology journal. And yes, this venue is on rather AI but we still should maintain some sensible quality on experiments if we are to show that. Currently, it just serves only as good as an anecdote.  (Question)


- It is shown to have much better performance against OpenAI's o1 model, but which DeLLma is used is not clear. (Question)

-  Organisation wise  the DELLMa main picture (there is no name unfortunately for the figure btw to refer to it, but perhaps understandable) comes before the explanations in 3.1 and other sections and create questions immediately. This is a very minor issue.

-Unclarities in  Page 9, Main results, second paragraph is not satisfying:  The vagueness of explanation is not a weakness per se since authors are careful to call these as hypotheses. Yet, the difference on phenomenon is very small, and now we enforce ourselves to explain the phenomenon, and likely we hallucinate ourselves.  The result can be much more simple: the amount of data considered, and the processing capacity of the model "can be" pretty much be  a reason that top 1, simplifies it much better compared to other.   Also see that when number of alternatives is 5 it already performs better, which might be due to a particular alternative. Relatedly the wayt the average is taken matters: 1) some difficulties might be due to assessing particular actions (are these scores are evaluated permuting through all the alternatives? (E.g., which 2 actions matters)?) 2) averaging across the accuracy score of  different size of scores matters. There I would go for higher weight to the higher number of actions.  These subtleties are unfortunately not clear, (question)

-Mini typo: availble fruits.

**Questions:**

See the points above as "(question)". Happy to read your take.

---

> ### Author Response · Authors · 2024-11-22
> **Official Comment by Authors 1/2**
>
> Thank you for your careful review and constructive comments! We are glad that you found DeLLMa to be interesting while grounded in classical decision theory. We hope to address your questions and concerns below.
>
> ### __Missing Details on Human Annotation Study__
>
> We appreciate your criticism and apologize for any confusion this may have caused! In our global response, we have provided the annotation guidelines and updated annotation results. In short, we have solicited a total of 412 annotations, 200 of which are annotated by 5 external volunteers. From these, we obtained an annotator-LLM agreement rate of 66.75% and an inter-annotator agreement rate of 67.0\% ($\pm$ 6.34\%). All annotation results and evaluation scripts have been updated to our supplementary material (under `DeLLMa-additional-annotation`).
>
> ### __Strong Assumption on K Independent Factors__
>
> We agree that this is indeed a strong assumption. In our initial experiments, we opted for this assumption due to its simplicity and its reasonable end-to-end performance in practice. But future works can indeed consider a more meaningful probability factorization (e.g., a probabilistic graphical model). Our additional calibration results indicate reasonable forecasting performance, which we discuss next.
>
> ### __Additional Analysis of Calibration Performance__
>
> Thank you for pointing this out. We provide additional details and reference values for our evaluation of the forecasting distribution (i.e., Section 4.1, Table 1).
>
> In Table 1, we aim to show the calibration performance of our LLM-based forecasting method. To contextualize the quantitative statistics reported in this table, we additionally report two sets of ECE statistics:
>
> - ECE under the uniform distribution forecast (ECE-Uniform).
> - ECE under the LLM forecast, but with random choice as the ground truth (ECE-Random). We report the standard deviation from 100 random trials.
>
> This yields the table below. Together, these results indicate that our state forecasting algorithm attains much improved performance in comparison with these baselines. We will upload code to reproduce these results as additional supplementary materials.
>
> |                                    | ECE  | ECE-Random  ($\pm$SD) | ECE-Uniform |
> |------------------------------|--------------------------|---------------------------|---------------------------|
> | DeLLMa (GPT-4)        | 0.062                       | 0.135 ± 0.092                   | 0.333
> | DeLLMa (Claude 3)    | 0.142                       | 0.157 ± 0.084                   | 0.333
> | DeLLMa (Gemini 1.5) | 0.064                       | 0.113 ± 0.083                   | 0.333
>
>
> Note that, to calculate the ECE, for a fixed set of latent factors (e.g., fruit price change, climate conditions), we manually find and annotate ground truth values — using the USDA report and web search. We can then compute metrics such as the expected calibration error (ECE) for forecasts of these latent factors given by a model.
>
> ### __Fixed Number of Alternatives__
>
> This is a great point—indeed, many real-world decisions do not attain a predefined set of alternatives, and this is an important future direction that we aim to tackle. Nonetheless, there are many important problems that are endowed with a fixed action set. For example, in healthcare, physicians often make decisions from a finite set of treatment options based on a patient’s symptoms and diagnostic result [2]. Similarly, in supply chain management, decision-makers regularly work within a bounded set of distribution strategies and inventory policies, such as selecting from fixed shipping options or standard order quantities [3]. With a focus on this setting, DeLLMa is our first step towards a more comprehensive system for real-life decision making.

---

> ### Author Response · Authors · 2024-11-22
> **Official Comment by Authors 2/2**
>
> ### __Improving Stock Result Analysis__
>
> We appreciate your detailed feedback! Our intention was indeed to provide an initial hypothesis for this phenomenon rather than a definitive explanation, and we agree that the better performance of DeLLMa-Top1 may stem from our dataset or model processing constraints.
>
> > (1) some difficulties might be due to assessing particular actions (are these scores evaluated permuting through all the alternatives? (e.g., which 2 actions matters)?)
>
> Yes, these scores are evaluated through all possible permutations of the alternatives.
>
> > (2) averaging across the accuracy score of different sizes matters. There I would go for higher weight to the higher number of actions.
>
> Thanks for this great suggestion! In our current version, we opted for unweighted average accuracy as our performance metric due to its simplicity, but we agree that assigning higher weights to more complex tasks is a sensible approach. We also recognize the importance of evaluating decision-making capabilities across different levels of task complexity. To this end, we are developing a follow-up benchmark that will incorporate stratified analysis to better assess performance across various complexities.
>
> Based on your feedback, we have revised our phrasing in this paragraph, which reads as follows:
>
>
> ```
> Additionally, DeLLMa-Top1 performs better than DeLLMa-Pairs in the stocks data. This difference may stem from the simplicity of utility elicitation from only the top action choice, which requires less data processing and may mitigate noise compared to enumerating all state-action pairs. We hypothesize that in high-volatility data like stocks, LLMs may struggle with pairwise comparisons due to potential hallucination issues, particularly when the model attempts to rank options without a clear ground truth. By focusing on the top choice, DeLLMa-Top1 could avoid accumulating noise from these internal rankings, achieving better performance.
> ```
>
> ### __DeLLMa vs o1__
>
> Apologies for the confusion. In Table 3, we are comparing our largest DeLLMa variant, with a _per action sample size_ of 64 and an _overlap percentage_ of 25\% against OpenAI o1. While smaller DeLLMa instantiations can outperform o1 (shown in Figure 3), we used our large variants as they attain similar costs (Line 519).
>
> ### __Writing Organization__
>
> Thank you for your thoughtful feedback on our paper structure! Due to page limitations, we had to reference the appendix for certain details, which may have affected readability. We’ll do our best to address this. Specifically, we’ll explore ways to consolidate key explanations within the main text and adjust figure placements to improve clarity.
>
> [1] Approaching Human-Level Forecasting with Language Models, https://arxiv.org/abs/2402.18563
>
> [2] MIMIC-IV-Ext Clinical Decision Making: A MIMIC-IV Derived Dataset for Evaluation of Large Language Models on the Task of Clinical Decision Making for Abdominal Pathologies, https://physionet.org/content/mimic-iv-ext-cdm/1.1/
>
> [3] Large Language Models for Supply Chain Optimization, https://arxiv.org/abs/2307.03875

---

> > ### Author Response · Authors · 2024-11-25
> > **Thanks for Your Valuable Input**
> >
> > Dear Reviewer RA2F,
> >
> > Thank you once again for your insightful feedback! We hope our response has addressed your concerns satisfactorily, and we would be happy to engage further if you have any additional questions or suggestions.

---

### Author Response · Authors · 2024-11-22
**General Response**

We first want to express our gratitude to all reviewers for their helpful feedback. All reviewers agree that our method is original while grounded in decision theory, our writing is clear, and experimental results show clear improvements from baseline methods. We first address a common question below:

### __Additional Details on Human Annotation Study (RA2F, 4x55)__:

We provide details on our human annotation guideline and new results. These results corroborate our initial findings reported in Table 4.

__Annotation Guidelines__: At each round, all human annotators are presented with a pair of LLM-ranked state-action tuples, denoted as $(s, a)_1$ and $(s, a)_2$. These tuples are presented in a random order to eliminate positional bias during the annotation procedure. Then, the annotators are asked to evaluate whether $(s, a)_1$ is more preferable to $(s, a)_2$, or vice versa. All our experiments are conducted on the agriculture dataset.

In addition to the paper authors, we have now added __200__ new annotations from 5 volunteers, totalling a set of __412__ pairwise comparisons across 3 LLM backbones. Our annotation results—measured as the average agreement rate between human and LLM preferences—are reported below:

|                                    | GPT-4  ($\pm$SD) | Claude 3  ($\pm$SD) | Gemini 1.5  ($\pm$SD) |
|-----------------------------------|--------------------------------|--------------------------------|--------------------------------|
| __Agreement \% with Human__    | 68.4\% ($\pm$ 3.6\%) |  65.3\% ($\pm$ 5.6\%)  |   65.7\% ($\pm$ 3.7\%)          |

Overall, we observe that LLM-human agreement is statistically significantly better than chance (50\%). Further, across the shared annotations, we observe an _inter-annotator_ agreement of __67.0\% ($\pm$ 6.3\%)__, which is on par with the human-LLM agreements.

All annotation results are available in our supplementary material under `DeLLMa-additional-annotation`

---

### Meta-Review · Area_Chair_zadV · 2024-12-18

**Metareview:**

The reviewers were generally positive about the paper, specifically with the framework and the rigor in the writing. There are suggestions toc make the paper self-contained and not refer so much to the appendix, increase the details on the experiments and justification of the inherent assumptions in the framework.  We hope the authors consider the feedback in improving the paper for the final version.

**Additional Comments On Reviewer Discussion:**

The consensus was positive.

---

### Decision · Program_Chairs · 2025-01-22

Accept (Spotlight)